# SFT or RL? An Early Investigation into Training R1-Like Reasoning Large Vision-Language Models

**Hardy Chen**[1]*, **Haoqin Tu**[1]*, **Fali Wang**[3], **Hui Liu**[4], **Xianfeng Tang**[4], **Xinya Du**[2], **Yuyin Zhou**[1], **Cihang Xie**[1]

[1] **University of California, Santa Cruz**   [2] **University of Texas at Dallas**
[3] **The Pennsylvania State University**   [4] **Amazon Research**

🌐 **Project Page:** `https://ucsc-vlaa.github.io/VL-Thinking/`

🤗 **7B Model:** `https://huggingface.co/UCSC-VLAA/VLAA-Thinker-Qwen2.5VL-7B`

🤗 **3B Model:** `https://huggingface.co/UCSC-VLAA/VLAA-Thinker-Qwen2.5VL-3B`

🤗 **Dataset:** `https://huggingface.co/datasets/UCSC-VLAA/VL-Thinking`

**Reviewed on OpenReview:** *https://openreview.net/forum?id=wZI5qkQeDF*

## Abstract

This work revisits the dominant supervised fine-tuning (SFT) then reinforcement learning (RL) paradigm for training Large Vision-Language Models (LVLMs), and reveals a key finding: on visual math reasoning tasks, SFT can significantly undermine subsequent RL by inducing "pseudo reasoning paths" imitated from expert models. While these paths may resemble the native reasoning paths of RL models, they often involve prolonged yet less informative reasoning steps. To systematically study this effect, we introduce `VLAA-Thinking`, a new multimodal dataset designed to support reasoning in LVLMs. Constructed via a six-step pipeline involving captioning, reasoning distillation, answer rewrite and verification, `VLAA-Thinking` comprises high-quality, step-by-step visual reasoning traces for SFT, along with a more challenging RL split from the same data source. Using this dataset, we conduct extensive experiments comparing SFT, RL and their combinations. Results show that while SFT helps models learn reasoning formats, it often locks aligned models into imitative, rigid reasoning modes that impede further learning. In contrast, building on the Group Relative Policy Optimization (GRPO) with a novel mixed reward module integrating both perception and cognition signals, our RL approach fosters more genuine, adaptive reasoning behavior. Notably, our model VLAA-Thinker, based on Qwen2.5VL 3B, achieves the best performance across six popular visual math reasoning benchmarks among 4B scale LVLMs, surpassing the previous state-of-the-art by 1.8%. We hope our findings provide valuable insights in developing reasoning-capable LVLMs and can inform future research in this area.

## 1 Introduction

Large Language Models (LLMs) with strong reasoning capability have recently gained wide attention with the emergence of OpenAI's o1/o3 and Deepseek-R1 (Guo et al., 2025; Jaech et al., 2024). A common practice to empower models with reasoning abilities comprises two steps: supervised fine-tuning (SFT) on reasoning data, followed by reinforcement learning (RL) to further boost performance. This successful paradigm has inspired efforts to extend these strengths beyond textual domains to Large Vision-Language Models (LVLMs) (Peng et al., 2025; Chen et al., 2025a; Deng et al., 2025b; Shen et al., 2025; Yang et al., 2025b).

---

*Equal contribution.

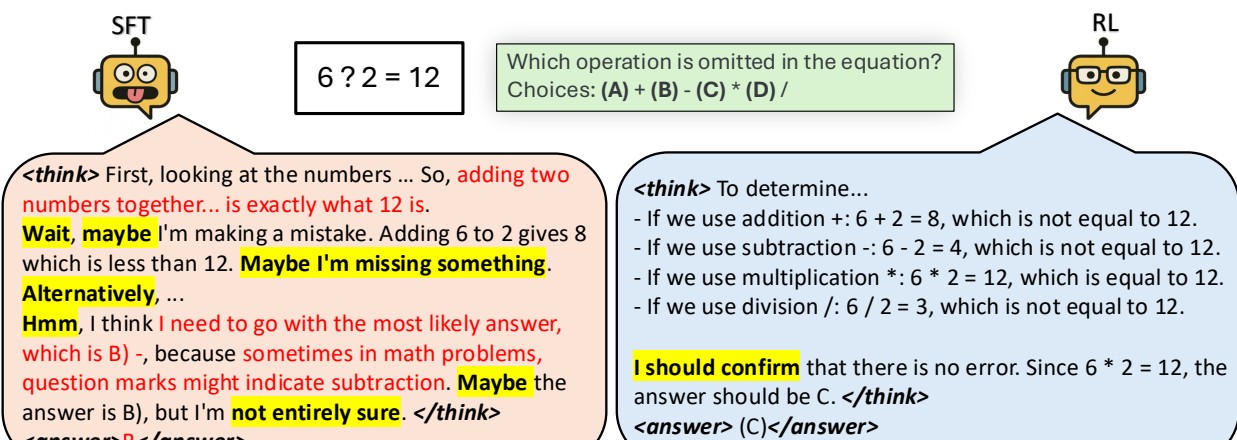

Figure 1: Examples from LVLMs trained with different strategies for reasoning. **Left**: response from a model trained with reasoning-SFT, showing *pseudo reasoning traces* and a number of *pseudo self-reflective cues* (*i.e.*, aha-moments) imitated from R1. **Right**: response from a model trained with RL, showing *native reasoning ability* and *authentic aha-moments* emerged from RL training. Wrong reasoning steps are colored red and aha-moments are highlighted.

In this work, we take a step further and examine whether the widely adopted "SFT then RL" paradigm similarly benefits the development of reasoning-capable LVLMs. Specifically, we ask: ***1) What are the distinct effect of reasoning-SFT[1] and RL on visual math reasoning? and 2) Is this two-stage paradigm truly necessary for visual math reasoning in LVLMs?*** To systematically explore these questions, we curate `VLAA-Thinking`, the first comprehensive and high-quality image-text reasoning dataset explicitly designed to support both SFT and RL. Unlike prior datasets, `VLAA-Thinking` includes detailed, step-by-step reasoning traces derived from the R1-style "think-then-speak" intermediate reasoning. We construct a dedicated SFT split featuring multimodal chain-of-thought (CoT) examples suitable for visual instruction tuning, alongside a more challenging RL split curated from the same source to encourage deeper and more adaptive reasoning behaviors. To effectively transfer reasoning capabilities from text-only models to the multimodal domain, we construct our dataset through a six-stage pipeline: metadata collection, image captioning, R1-based distillation, answer rewriting, verification, and split curation. Specifically, we input image captions and visual questions into DeepSeek-R1 to generate initial reasoning traces. These outputs are then rewritten for improved fluency and verified for correctness using a GPT-based verifier, resulting in high-quality multimodal reasoning dataset for SFT and RL.

Next, we carefully ablate the role of SFT, RL and their combinations in multimodal reasoning using our `VLAA-Thinking` dataset. To better understand the role of SFT, we perform a detailed analysis, systematically examining the impact of SFT data type (*e.g.*, with and without the self-reflective "aha moments"), dataset scale, and model capacity. To explore the potential of RL in the vision-language context, we design a novel mixed reward function within the Group Relative Policy Optimization (GRPO) (Shao et al., 2024) framework that involves both perception and cognition rewards to incentivize the model to produce well-reasoned answers. Specifically, our mixed reward signal blends 2 types of reward with 5 types of functions. For **rule-based questions**, there are functions for *digit*, *multiple-choice*, *math* and *bounding box* outputs. For **open-ended questions**, we adopt a competent reward model, *XComposer-2.5-RM* (Zang et al., 2025), along with a reference-based reward method to score an answer. We then closely investigate the effects of different reward functions, base models, and the interaction between SFT and GRPO to further optimize reasoning capabilities.

Our extensive experiments comparing SFT and RL reveal several noteworthy insights. First, we probe the contribution of SFT and RL in multimodal reasoning: while SFT improves performance on standard tasks

---

[1]We use SFT to represent reasoning-SFT hereinafter for a more concise terminology. SFT is used to endow models with various abilities (*e.g.*, basic instruction following ability which a pretrained base model lacks) while reasoning-SFT is used to specifically introduce reasoning patterns to models.

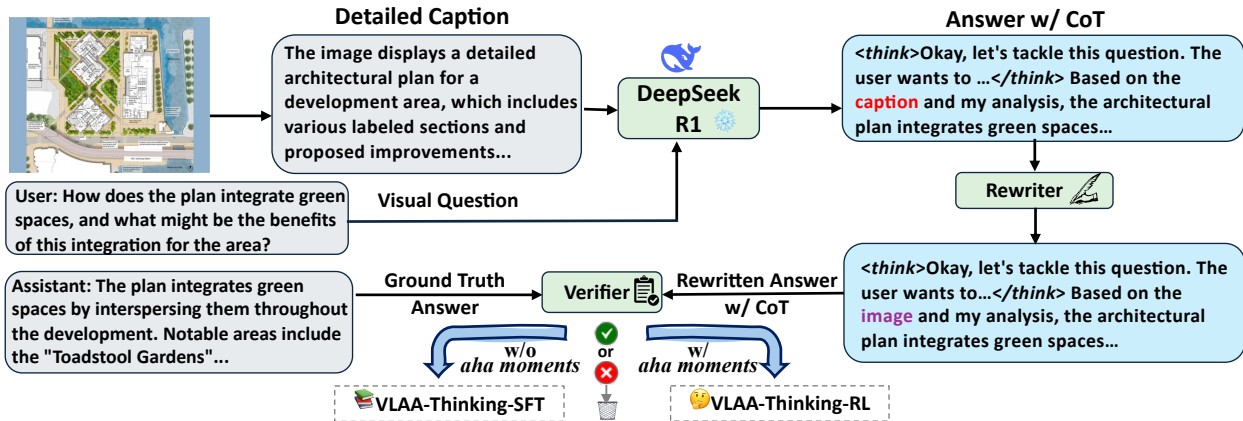

Figure 2: **Data generation pipeline.** We first generate initial reasoning traces by feeding detailed captions and visual questions into DeepSeek-R1.These outputs are then rewritten for improved fluency and verified for correctness using a GPT-based verifier. the resulting data is split into `VLAA-Thinking-SFT` and `VLAA-Thinking-RL`.

over the base model, it falls short in enhancing complex reasoning. Merely imitating an expert's thinking through SFT often induces "*pseudo reasoning paths*", a superficial reasoning pattern which may contain "*pseudo aha moments*" (superficial self-reflective cues), as illustrated in Figure 1. We show that these imitated reasoning patterns can hinder genuine reasoning advancement, *i.e.*, 47% relative performance drop on 7B models. This observation is also in line with recent studies highlighting the need for feedback and exploration signals to drive advanced reasoning behaviors (Peng et al., 2025). Additionally, our ablations show that for rule-based rewards, math and multiple-choice are more beneficial than others, and that a combination of both rule-based and open-ended rewards yields the best performance.

While prior work suggests that SFT followed by RL in LVLMs offers the best of both worlds (Guo et al., 2025; Yang et al., 2025b; Deng et al., 2025b)—first mimicking good reasoning format, then refining via RL feedback, we find that **applying SFT before GRPO *hurts* performance on aligned models**, with an average 12.7% drop, and even a smaller scale SFT leads to a similar decline. Regarding model size, larger models cannot immune from the degeneration brought by SFT, as 7B models share almost the same performance drop with their smaller counterparts. Finally, examining the training procedure, we observe little correlation between response length, reward, and performance—SFT-ed models get higher initial rewards and longer response yet underperform RL-trained ones, contrasting with the previous observation that better models usually produce longer answers with higher RL reward (Guo et al., 2025; Peng et al., 2025).

To summarize, while SFT helps unaligned models follow instructions, it limits exploration during RL by promoting imitative reasoning. In contrast, learning directly from reward signals yields more effective and adaptable thinking behavior. Empirically, direct RL proves superior. Our model, **VLAA-Thinker-Qwen2.5VL-3B**, achieves the **top-1** performance on the Open LMM Reasoning Leaderboard among 4B-scale LVLMs, surpassing the previous state-of-the-art by 1.8%. Our case study further emphasizes these gains with more concise, effective reasoning traces presented in model answers.

## 2 The `VLAA-Thinking` Dataset

To systematically evaluate the "SFT then RL" paradigm for developing reasoning capabilities in LVLMs, we construct `VLAA-Thinking`, a dataset that consists of two parts: 1) `VLAA-Thinking`-SFT which captures step-by-step reasoning grounded in visual inputs for SFT, and 2) `VLAA-Thinking`-RL which contains challenging samples designed specifically for RL. Our data generation pipeline is designed to transfer reasoning capabilities from a powerful text-only model to the multimodal domain through a structured, multi-stage process. The entire pipeline, as illustrated in Figure 2, consists of six key components:

| Name | Data Type | #Ori. | #Pipeline | #Final SFT | #Final RL |
|---|---|---|---|---|---|
| *Collected from Distilling R1* | | | | | |
| CLEVR-Math | Closed-end | 35,000 | 28,018 | 5,923 | 2,000 |
| GeoQA170K | Closed-end | - | - | - | 6,499 |
| Math PUMA | Closed-end | 30,000 | 26,672 | 19,258 | 6,696 |
| ArxivQA | Closed-end | 54,399 | 51,348 | 34,604 | 1,000 |
| DocVQA | Closed-end | 10,194 | 8,206 | 4,897 | 1,000 |
| VizWiz | Closed-end | 20,523 | 6,528 | 4,266 | 1,000 |
| ALLaVA-LAION | Open-end | 47,066 | 18,123 | 10,496 | 3,000 |
| *Collected from LLaVA-CoT* | | | | | |
| COCO | Closed-end | 3,000 | 3,000 | 8,727 | 2,000 |
| VisualGenome | Closed-end | 3,000 | 3,000 | 38,242 | 2,000 |
| Total | Closed- & Open-end | 203,182 | 144,895 | 126,413 | 25,195 |

Table 1: **Data statistics of `VLAA-Thinking`.** We present the original volume of metadata (#Ori.), the data size after the distillation pipeline (#Pipeline), the size of sampled examples for SFT (#Final SFT) and RL (#Final RL), respectively. Note that we only use GeoQA170K with verifiable answers for the RL split.

**#1: Metadata Collection**   We collect metadata from 9 vision-language datasets featuring either closed- or open-ended questions. Specifically, we sample data containing unique images from CLEVR-Math (Lindström & Abraham, 2022), Math PUMA (Zhuang et al., 2024), ArxivQA (Li et al., 2024b), DocVQA (Mathew et al., 2021), VizWiz (Gurari et al., 2018), and ALLaVA (Chen et al., 2024a), and process them through our complete data pipeline. In addition, we directly adopt COCO and VisualGenome data from LLaVA-CoT (Xu et al., 2024). An exception is GeoQA170K (Gao et al., 2023), which we include only in the RL split due to persistent hallucination issues during captioning. Detailed statistics are in Table 1.

**#2: Visual Input and Additional Information**   Each sample begins with an image, question, and its corresponding answer. To bridge the gap between the visual modality and language reasoning, we resort to GPT-4o to generate a detailed image caption describing the content in structured and semantically rich language (detailed prompts in Appendix A.1). During this process, we take full advantage of the provided knowledge in the data beyond just the GPT captions. In detail, we provide these dataset-specific information: (1) CLEVR-Math: Instructions for synthesizing the image from CLEVR (Johnson et al., 2017); (2) Math PUMA: Textual description of math problems in the image from the dataset itself. (3) ALLaVA-LAION: Fine-grained and verified GPT-4V captions from the original dataset.

**#3: Reasoning Answer Distillation**   We utilize a strong text-only reasoning model: DeepSeek-R1 to generate thinking rationale and final answers. The model is provided with the image caption, the visual question, and additional information from certain datasets. It responds using a structured reasoning format that is between `<think>` and `</think>` tags and contains a sequence of logical steps leading to the final answer.

**#4: Answer and Rewriting**   To enhance consistency and eliminate modality-specific artifacts, the raw reasoning answers generated by R1 are passed through a rewriting module (*i.e.*, GPT-4o (OpenAI et al., 2024) in our experiment). This module removes unnecessary phrases (*e.g.*, references to "caption"), and ensures the answer adheres to a clean, instruction-following format based on the image. We further filter out samples with the sentence length gap larger than 15 words to ensure minimum modifications in this process.

**#5: Automated Verification**   To assess whether the generated reasoning answers is correct regarding the groundtruth answer, we implement an automated verifier. This verifier compares the rewritten reasoning answer to the groundtruth of the visual question, determining whether the outputs are correct or incorrect. Only the examples that are verified as correct are retained as the final training data.

**#6: Curating Splits for SFT and RL**   The last step of our data generation pipeline is to curate two non-overlapped training sets for SFT and RL, respectively. Inspired by Chu et al. (2025) which finds that RL is particularly effective in encouraging deeper reasoning on challenging cases, we aim to select more challenging samples for the RL split. To achieve this, we propose using the presence of *self-reflective cues* (*i.e.*, the "aha moments") in the distilled answers as an indicator of a sample's difficulty level (details are in

Appendix A.2). For the SFT split, we exclude samples *with "aha moments"*, as such samples may be too complex to fully imitate through finetuning. On the other hand, the harder examples with "aha moments" form the RL split, on which reward-driven learning may be better suited to elicit meaningful reflection.

Following these steps, our dataset adheres to the format {image, question, reasoning, answer}, with reasoning and answer generated by DeepSeek-R1. We construct a high-quality multimodal reasoning dataset with 126,413 samples for SFT and 25,195 samples for RL. We provide results of quality evaluation in Appendix A.4.

# 3 Investigating The Role of SFT for Multimodal Reasoning

SFT has become the de-facto approach for training LLMs. Recent studies aim to extend the strengths of SFT to empower LVLMs with reasoning abilities by training on specially formatted data.Unlike prior methods that incorporate standalone textual descriptions of images (Xu et al., 2024), this direct strategy enables the model to develop grammatically coherent reasoning abilities, allowing it to "think before speak." In recent vision-language reasoning systems, there is a notable trend of complementing or even replacing SFT with RL to enhance complex reasoning abilities (Peng et al., 2025; Deng et al., 2025b). We follow this line and take it further by probing the underlying cause of this shift. Our finding suggests that self-reflection thinking ("aha moments") from the SFT process is overloaded with excessive and irrelevant reasoning, becomes what we call "pseudo aha moments" and ultimately hurts performance. In this section, we explore **1)** the model perform when SFT-ed on data with aha-moments and **2)** the effect of SFT data size to model performance.

## 3.1 Experiment Setup

To investigate the effect of SFT training with aha-moments, we collect the distilled VQA pairs whose distilled answers contain aha-moments, totaling 55K samples. To study the effect of SFT with different sizes of training sets, we use perplexity (PPL) filtering to obtain a smaller SFT dataset. Specifically, we compute the PPL score of each answer in `VLAA-Thinking-SFT-126K` using Qwen2VL-2B and Qwen2.5VL-3B, and sort all samples by their average PPL scores over the two models. We keep the samples with high PPLs to obtain a total of 25K SFT samples, as these harder examples push models to learn more effectively and efficiently (Ankner et al., 2024; Li et al., 2024c).

We select four models for training: Qwen2VL (2B and 7B)[2], Qwen2.5VL (3B and 7B). Each model is trained with a batch size of 128 and their vision encoder frozen. We evaluate model performance with VLMEvalKit (Duan et al., 2024) on 6 math reasoning benchmarks hosted in Open LMM Reasoning Leaderboard, which contains 6 challenging math reasoning benchmarks including MathVista (Lu et al., 2024), MathVision (Wang et al., 2024b), MathVerse (Zhang et al., 2024), DynaMath (Zou et al., 2024), WeMath (Qiao et al., 2024), LogicVista (Xiao et al., 2024). We present the percentage of relative performance drop of different models in Figure 3. Detailed training and evaluation setup are in Appendix B.

## 3.2 Findings

**SFT with *Aha Moments* Degrades Performance.** We present results for the Qwen2.5VL-3B model trained under three different settings using our SFT data in Table 2. Somewhat unexpectedly, the model finetuned on 55K examples containing the *aha moment* performs significantly worse than the base model, with an average drop of 10.5%. This suggests that chasing the *aha moment* through SFT is unreliable, as SFT merely teaches the model to mimic rather than to generalize genuine self-reflective reasoning. Additionally, the table shows evidence that straightforward SFT using multimodal reasoning data also degrades performance, *e.g.*, we observe an average drop of 10.2% and 19.1% when fine-tuning on 25K and 126K samples, respectively.

| Model | Avg. |
|---|---|
| Qwen2.5VL-3B | 31.8 |
| w/ *aha*-55K | 21.3 |
| w/ 25K | 21.6 |
| w/ 126K | 12.7 |

Table 2: Average performance over 6 reasoning benchmarks of Qwen2.5VL-3B SFT-ed on different sizes of SFT data and on data containing only examples with aha moment (*aha*-55K).

---

[2]In this work, Qwen2VL-2B and Qwen2VL-7B refer to the instruction-tuned versions.

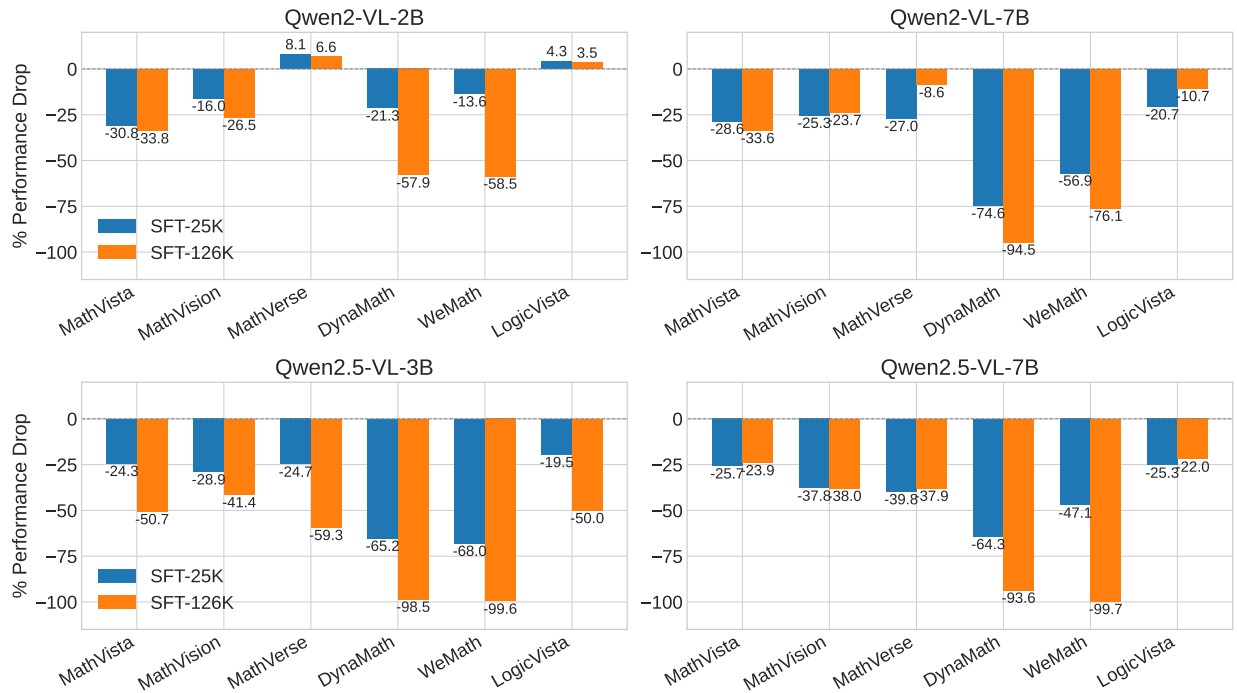

Figure 3: **Delta percentage performance change** of different models trained with supervised fine-tuning (SFT) only.

**More SFT Data, Worse Performance.** Counterintuitively, even a five-fold increase in the supervised dataset (from 25K to 126K instances) often fails to improve performance and in most cases actually *harms* it. Models trained with 126K SFT samples suffer a relative performance drop of over average 14% compared to their 25K-trained counterparts over all model and task settings (*e.g.*, 25K: 32.2% *vs.* 126K: 47.0%). This degradation is particularly evident on complex datasets such as WeMath and DynaMath, where the relative decrease reaches as high as 97.9% over Qwen2.5VL models on average. Even on mid-difficulty benchmarks like MathVision and MathVerse (*i.e.*, model performance is relatively higher), the 126K SFT models underperform, with an average drop of 28.6% compared to the untrained model over 4 models. These results suggest that simply scaling up SFT data does not boost generalizable reasoning skills of LLMs, and may instead suppress the model's capacity on various reasoning tasks.

**Larger Models Are Not Immune to SFT Degeneration.** Contrary to expectations, scaling up model size does not mitigate the adverse effects of excessive SFT, under heavier SFT they exhibit pronounced drops on the most challenging evaluations. A larger 7B models fine-tuned on 126K examples experience drops nearly identical in magnitude to their smaller 2B or 3B counterparts: 47.2% for smaller models *vs.* 45.4% for larger models compared with base models. Notably, despite the strong performance of Qwen2.5VL-7B model (*e.g.*, 68.1% on MathVista), it also suffers an average decline of 52.5% on these reasoning tasks when SFT-ed with 126K data.

These findings highlight the limitations of SFT as a tool for enhancing multimodal reasoning. While it may be suitable for learning reasoning formats, it falls short of the expectations for fostering inherent self-reflection. Rather than simply scaling supervision data, our results suggest for a shift toward more advanced training methods like RL.

# 4 Improving Multimodal Reasoning with Mixed Rewards

The previous section shows that SFT is insufficient to transfer R1's ability to LVLMs on vision-language tasks. Therefore, it is crucial to seek for other post-training methods to elicit the reasoning ability of LVLMs. Since reinforcement learning (RL) is effective in enhancing reasoning ability (Yang et al., 2025a; Kirk et al.,

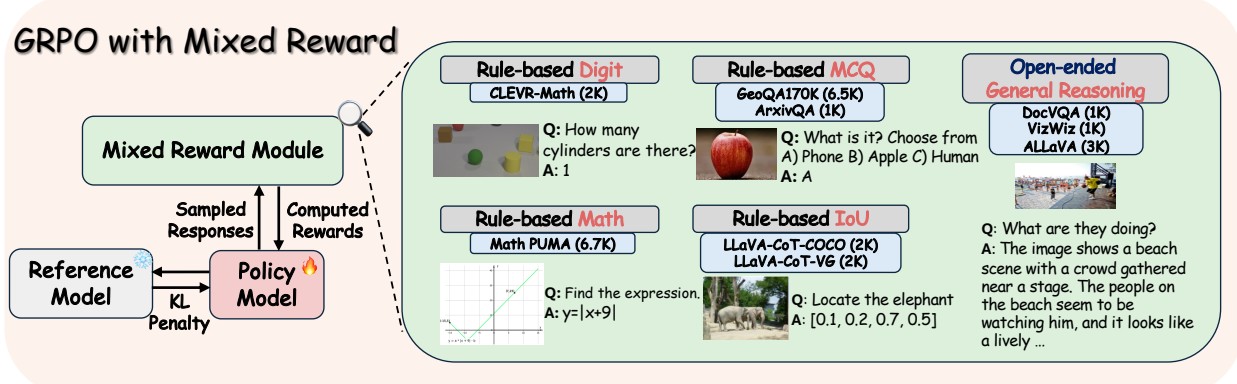

Figure 4: The proposed **Mixed Reward Module** for GRPO training, comprising 2 reward formats (rule-based and open-ended) and 5 types of verifiable rewards (digit, MCQ, math, IoU and general reasoning).

2023), and GRPO has recently been proven more effective and efficient on textual math reasoning task (Shao et al., 2024; Jahin et al., 2025) than other methods like PPO (Schulman et al., 2017), it motivates us to apply GRPO training for vision-language reasoning tasks.

Mathematically, let $q$ be a query and $\{o_i\}_{i=1}^G$ be a group of $G$ sampled outputs from the old policy model $\pi_{old}$, GRPO maximizes the following objective:

$$\mathcal{J}_{\text{GRPO}}(\theta) = \mathbb{E}_{q, \{o_i\} \sim \pi_{\theta_{\text{old}}}} \left[ \frac{1}{G} \sum_{i=1}^G \frac{1}{|o_i|} \sum_{t=1}^{|o_i|} \min \left( r_t(\theta) \hat{A}_{i,t}, \text{clip}(r_t(\theta), 1 - \epsilon, 1 + \epsilon) \hat{A}_{i,t} \right) \right] - \beta D_{\text{KL}}(\pi_\theta \,\|\, \pi_{\text{ref}})$$

and

$$r_t(\theta) = \frac{\pi_\theta(o_{i,t}|q, o_{i,<t})}{\pi_{\theta_{old}}(o_{i,t}|q, o_{i,<t})}$$

where $\hat{A}_{i,t}$ is the estimated advantage, $\beta$ is the KL penalty coefficient and $\pi_\theta$, $\pi_{\theta_{\text{old}}}$, $\pi_{\text{ref}}$ are current, old, and reference policies, respectively.

## 4.1 GRPO with Mixed Reward

To better adapt GRPO to multimodal reasoning, in addition to adopting the rule-based reward similar to the textual GRPO training, it is necessary to consider additional characteristics introduced by the vision modality. Inspired by (Fu et al., 2024) which benchmarks LVLMs by *perception* and *cognition* (reasoning), we propose a **mixed reward framework** for GRPO training, as illustrated in Figure 4. The reward system comprises five types of verifiable rewards with two formats, encompassing both visual perception and visual reasoning tasks.

**Rule-Based Reward** There are 4 types of rule-based rewards, including digit matching, option letter matching and math expression matching and Intersection over Union for bounding boxes. For digit matching, the model is asked to answer counting questions from CLEVR-Math whose groundtruths are a single digit. For option letter matching, the model is required to answer a multiple-choice question (MCQ). For math expression matching, the model is asked to solve a math question, such as finding a function expression or the volume of a cone, and output its answers in latex format. We use the `Math Verify`[3] package to check for correctness. For bounding boxes, the model is prompted to output the bounding box coordinates of an object in the image, and an IoU score (range from 0 to 1) is computed as reward.

**Open-ended Reward** We leverage InternLM-XComposer2.5-Reward (Zang et al., 2025) as the scorer, denoted as $S_\theta(\cdot)$, which takes an image and a QA pair as input, and outputs a reward score. Following Muhtar

---

[3]https://github.com/huggingface/Math-Verify

et al. (2025), the reward for a sampled response $\hat{y}$ is computed as $R_{open} = 1 - \exp(-(S_\theta(\hat{y}) - S_\theta(y)) \times \beta)$ if $f_\theta(\hat{y}) > f_\theta(y)$ else 0, where $S_\theta(y)$ is the score of the reference answer, and $\beta$ is a smoothing hyperparameter. Note that the open-ended reward is normalized into [0,1], which is consistent with the scale of rule-based reward, partially avoiding reward hacking during training.

**Implicit Format Reward**   Unlike Guo et al. (2025) and its subsequent works which use a separate reward term for format correctness, we discard this format reward term and make the format reward supersede all other rewards. Namely, whenever we are unable to extract a valid response from the raw answer, the reward would be 0. We empirically find that by specifying the output format in system prompt, the model is able to generate answers with correct formats through trials and errors. The implicit format reward design simplifies the reward computation. Further, it may yield better performance since less restriction is imposed on the exploration process (Zeng et al., 2025).

## 4.2   Effect of SFT on GRPO Training

| GRPO Backbone | MathVista | MathVision | MathVerse (vision-only) | DynaMath (worst) | WeMath | LogicVista | Avg. |
|---|---|---|---|---|---|---|---|
| Qwen2VL-7B-Inst | 59.6 | 19.8 | 33.9 | 15.2 | 30.5 | 36.0 | 32.5 |
| Qwen2VL-7B-Inst+SFT | 43.7 | 14.7 | 19.0 | 3.2 | 11.1 | 27.3 | 19.8$_{(-39\%)}$ |
| Qwen2VL-7B-Base | 59.3 | 18.2 | 33.5 | 11.4 | 23.2 | 36.2 | 30.7 |
| Qwen2VL-7B-Base+SFT | 49.5 | 16.4 | 25.0 | 6.4 | 20.4 | 32.7 | 25.7$_{(-16\%)}$ |

Table 3: **Benchmark results of models trained with GRPO on different backbones.** SFT+GRPO yields performance degradation, indicating that SFT is NOT compatible with GRPO in multimodal reasoning.

**SFT is NOT Compatible with GRPO in Multimodal Reasoning.**   Although we reveal in Section 3 that SFT alone leads to a performance drop in multimodal reasoning, it is still unclear whether SFT plays a crucial role in aiding GRPO, like the golden key in DeepSeek-R1. We experiment with different backbones for GRPO training. Specifically, we adopt Qwen2VL-7B-Base and Qwen2VL-7B-Inst, and perform SFT on them with 25K samples, followed by GRPO training.

From Table 3, we observe that models undergoing SFT before GRPO training perform worse than those trained with GRPO alone, presenting an average drop of 8.9% across Qwen2VL-Base and Qwen2VL-Inst compared to their non-SFT counterparts. We also find that SFT introduces more degradation to instruction models than to base models without instruction-following capabilities. For instance, Qwen2VL-Inst suffers a 7.7% more drop in performance than Qwen2VL-Base post-SFT, suggesting that SFT can compromise the instruction-following ability crucial for effective GRPO training. Taken together, these results suggest that SFT is currently incompatible with GRPO in the context of multimodal reasoning, impairing both base and instruction-tuned LVLMs.

**Smaller SFT Dataset Still Jeopardizes GRPO Performance.**   Since we reveal in Section 3.2 that more SFT data yields lower performance, we try to investigate the effect of downsizing the SFT training set. Following the PPL filtering method in Section 3, we select top-10K and top-5K samples from `VLAA-Thinking-SFT-126K` to finetune Qwen2.5VL-3B, followed by GRPO training. For comparison, we also conduct GRPO training without SFT.

We present the performance of Qwen2.5VL-3B on each task in Figure 5. A clear observation is that applying SFT on 5K examples prior to GRPO significantly degrades performance compared to using GRPO alone, showing an average drop of 13.5%. Moreover, scaling up SFT data to 10K yields only a marginal improvement of 0.8%. These results further support that SFT before GRPO can hinder the model's learning capability.

**Response Length, Reward, and Model Performance are NOT Necessarily Related.**   Prior work in RL suggests that longer responses often correlate with better reasoning and higher RL rewards (Guo et al.,

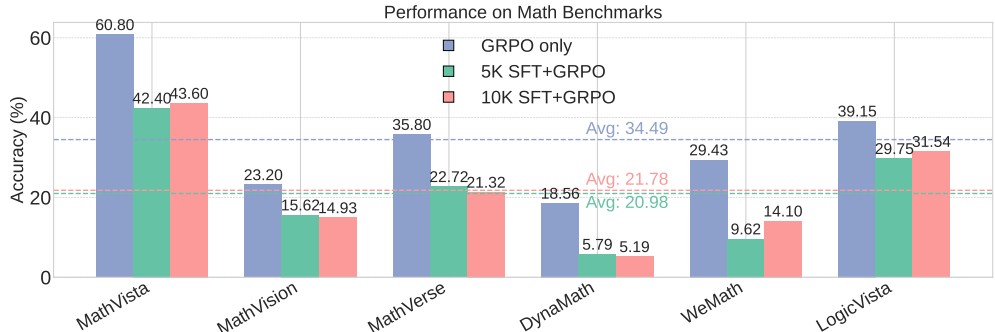

Figure 5: **Impact of SFT with 5K and 10K samples before GRPO.** Smaller-sized SFT datasets still jeopardizes GRPO performance.

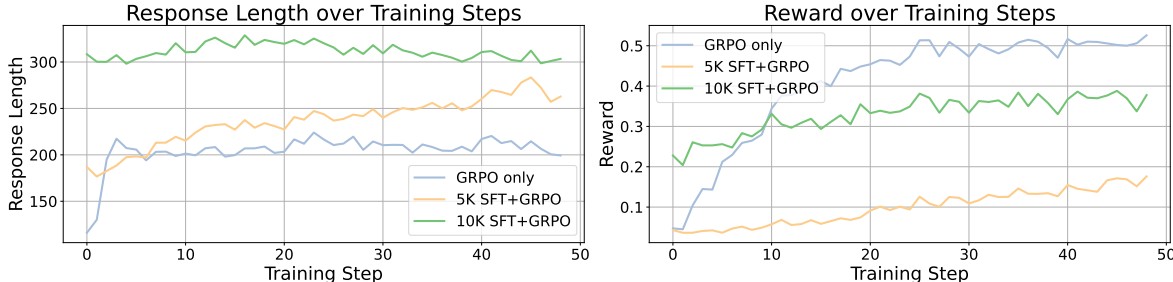

Figure 6: **Response length (left) and reward (right) during training.** Training with only GRPO yields the lowest response length and yet the highest final reward and best benchmark performance, indicating that response length, reward, and model performance are NOT necessarily related.

2025; Zhou et al., 2025; Chen et al., 2025c). However, our findings in Figure 6 reveal that response length and reward in GRPO are not reliable indicators of reasoning ability. For instance, the 10K SFT+GRPO model produces the longest responses but ends up with lower rewards than the GRPO-only model (∼0.35 vs. ∼0.5) after training. Similarly, the 5K SFT+GRPO variant shows moderate length and reward but still underperforms on downstream tasks.

Interestingly, both SFT-ed models start with higher initial rewards (*e.g.*, ∼0.20 for 10K SFT+GRPO *vs.* ∼0.05 for GRPO-only), which is likely due to their early learning experience with supervision since SFT and GRPO data share the same distribution. However, they exhibit limited reward improvement during training, whereas the GRPO-only model rapidly surpasses them. These trends further reveal that SFT solely provides a higher "lower bound" for RL training, yet it may lower the "upper bound" since the reasoning SFT data constrains the model's exploration paths. Therefore, **reasoning is a native emerging ability that is more likely to be developed through RL, not SFT**. While SFT-ed models may appear to reason, their behavior is closer to pattern imitation — **a form of pseudo-reasoning that lacks the generalizable reasoning skills**.

## 4.3 GRPO Training *without* SFT

**Training Setup**   Following the findings in the previous section, we directly conduct GRPO training which yields four models: VLAA-Thinker-Qwen2VL-2B, VLAA-Thinker-Qwen2VL-7B, VLAA-Thinker-Qwen2.5VL-3B, VLAA-Thinker-Qwen2.5VL-7B. We also train on a base model of Qwen2VL-7B, and the resulting model is named VLAA-Thinker-Qwen2-7B-Zero. We sample 4 times for each query with temperature 0.8. Rollout and training batch size are set as 512 and 256, respectively. We train our model for 1 episode (outer loop) and 1 epoch per episode (inner loop) on 8*H100 GPUs with 49 steps. More details of training setup are in Appendix C.1.

| Model | MathVista | MathVision | MathVerse (vision-only) | DynaMath (worst) | WeMath | LogicVista | Avg. |
|---|---|---|---|---|---|---|---|
| *4B-scale LVLMs* | | | | | | | |
| Qwen2VL-2B | 48.0 | 16.1 | 17.5 | 3.8 | 10.8 | 26.6 | 20.5 |
| Qwen2.5VL-3B | 61.2 | 21.9 | 31.2 | 13.2 | 22.9 | 40.3 | 31.8 |
| VLM-R1-Math-0305 | 62.7 | 21.9 | 32.2 | 13.0 | 30.0 | **40.5** | 33.4 |
| Taichu-VLR-3B | **64.9** | 23.1 | 32.1 | 12.6 | 30.4 | 38.7 | 33.6 |
| **VLAA-Thinker-Qwen2VL-2B** | 43.6 | 14.8 | 19.0 | 3.4 | 12.6 | 30.4 | 20.3 |
| **VLAA-Thinker-Qwen2.5VL-3B** | 61.0 | **24.4** | **36.4** | **18.2** | **33.8** | 38.5 | **35.4** |
| *7B-scale LVLMs* | | | | | | | |
| LLaVA-OneVision-7B | 58.6 | 18.3 | 19.3 | 9.0 | 20.9 | 33.3 | 26.6 |
| InternLM-XComposer2.5 | 64.0 | 17.8 | 16.2 | 8.2 | 14.1 | 34.7 | 25.8 |
| Qwen2VL-7B | 61.6 | 19.2 | 25.4 | 11.0 | 22.3 | 33.3 | 28.8 |
| Qwen2.5VL-7B | 68.1 | 25.4 | 41.1 | 21.8 | 36.2 | 47.9 | 40.1 |
| InternVL2.5-8B | 64.5 | 17.0 | 22.8 | 9.4 | 23.5 | 36.0 | 28.9 |
| InternVL3-8B | 70.5 | **30.0** | 38.5 | **25.7** | 39.5 | 44.5 | 41.4 |
| R1-OneVision | 65.4 | 22.7 | 41.2 | 15.0 | 20.8 | 42.1 | 34.5 |
| **VLAA-Thinker-Qwen2VL-7B-Zero** | 59.3 | 18.2 | 33.5 | 11.4 | 23.2 | 36.2 | 30.7 |
| **VLAA-Thinker-Qwen2VL-7B** | 59.6 | 19.8 | 33.9 | 15.2 | 30.5 | 36.0 | 32.5 |
| **VLAA-Thinker-Qwen2.5VL-7B** | 68.0 | 26.4 | **48.2** | 22.4 | **41.5** | **48.5** | **42.5** |

Table 4: Evaluation results of 6 math reasoning benchmarks on Open LMM Leaderboard. VLAA-Thinker models significantly outperform baselines and other models.

**Evaluation Setup**  We follow the identical evaluation setup as described in Section 3.1. For 4B-scale baselines, we have Qwen2VL-2B (Wang et al., 2024c), Qwen2.5VL-3B (Bai et al., 2025), VLM-R1-Math-0305 (Shen et al., 2025), and Taichu-VLR-3B[4]. For 7B-scale baselines, we have LLaVA-OneVision-7B (Li et al., 2024a), InternLM-XComposer2.5 (Zang et al., 2025), Qwen2VL-7B (Wang et al., 2024c), Qwen2.5VL-7B (Bai et al., 2025), InternVL2.5-8B (Chen et al., 2025b), InternVL3-8B (Zhu et al., 2025), and R1-OneVision (Yang et al., 2025b). We present evaluation results in Table 4 and list our main findings below.

**Direct GRPO Training Boosts Model Performance.**  Models trained directly with GRPO on the VL-Thinking RL consistently outperform their respective base models. For example, at the 7B scale, two models trained on VL-Thinking achieve an average score of 36.5%, marking a 2.0% improvement over their base model average of 34.5%. Moreover, our best-performing 7B model consistently outperforms other similarly sized LVLMs (*e.g.*, InternVL2.5-8B, LLaVA-OneVision-7B), while our 3B model surpasses the recent reasoning-focused model, VLM-R1-Math, by 1.1% on average. These results once again demonstrate that GRPO significantly enhances reasoning capabilities, even without additional SFT.

**Stronger Instruction Model Leads to Better Post-GRPO Reasoning.**  An interesting observation is that model with better instruction tuning generally performs better. The instruction-aligned Qwen2-7B model, after GRPO, outperforms its unaligned counterpart VLAA-Thinker-Qwen2-7B-Zero by 1.8% on average (31.3% *vs.* 29.5%), with notable gains on harder tasks like DynaMath (5.0%) and WeMath (3.1%). Moreover, using a stronger instruction-tuned model for GRPO further improves across both 3B and 7B scales — VLAA-Thinker-Qwen2.5 surpasses VLAA-Thinker-Qwen2 by 12.6% on average, confirming that higher-quality instruction tuning leads to more effective post-RL reasoning.

**Emergence of Authentic *Aha Moments*.**  To show that our GRPO training can induce authentic self-reflection process, we plot the frequency of four aha expressions ("alternatively", "double-check", "i should check", "wait") for each VLAA-Thinker model in Figure 7. Since all models are trained using GRPO without being SFT-ed on distilled reasoning paths, all aha moments emerge from the GRPO process, demonstrating the model's self-developed reflective ability. Another finding is that the number of *aha moments* is not directly correlate with overall model performance, as more *aha moments* do not necessarily translate to higher reasoning scores.

---

[4] https://docs.wair.ac.cn/taichu/introduction.html

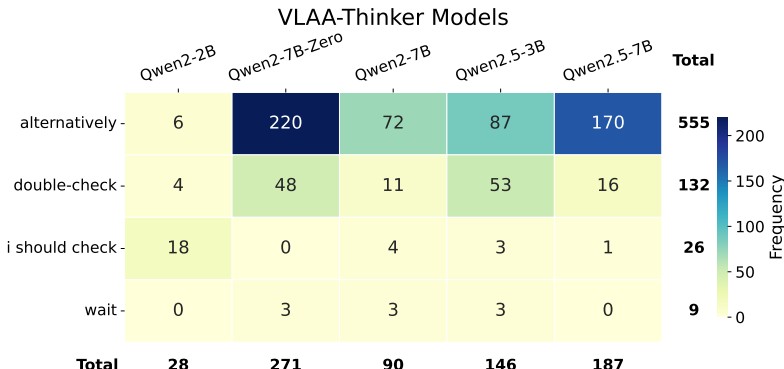

Figure 7: **Heatmap of different "aha" expressions** generated by VLAA-Thinker models during training.

## 4.4 Ablations

| Row | Method | Digit | Math | MCQ | IoU | Open-ended | MVi | MVs | WM |
|---|---|---|---|---|---|---|---|---|---|
| 0 | Qwen2.5VL-3B | | | | | | 21.9 | 31.2 | 22.9 |
| 1 | w/o Digit | | ✓ | ✓ | ✓ | | 23.5 | 34.6 | 28.8 |
| 2 | w/o Math | ✓ | | ✓ | ✓ | | 21.4 | 32.7 | 27.0 |
| 3 | w/o MCQ | ✓ | ✓ | | ✓ | | 21.5 | 33.9 | 18.4 |
| 4 | w/o IoU | ✓ | ✓ | ✓ | | | 22.8 | 35.3 | 30.0 |
| 5 | All Rule-Based | ✓ | ✓ | ✓ | ✓ | | 22.2 | 34.9 | 30.1 |
| 6 | **Mixed Reward** | ✓ | ✓ | ✓ | ✓ | ✓ | **24.4** | **36.4** | **33.8** |

Table 5: **Ablation of Mixed Reward** on MVi: MathVision, MVs: MathVerse and WM: WeMath. A combination of rule-based and open-ended rewards yields significant boost in performance.

**Mixed Reward.** To demonstrate the effectiveness of our mixed reward strategy, we perform an ablation study on Qwen2.5VL-3B by selectively disabling individual reward components and evaluating performance across three math reasoning benchmarks, as shown in Table 5. The model trained with **Mixed Reward** achieves the best overall performance, with an average improvement of 6.2% over the baseline, demonstrating the effectiveness of our reward design. Using only rule-based rewards (All Rule-Based) also yields consistent gains (*e.g.*, 29.1% vs. 25.3% baseline), while removing specific components—especially MCQ (w/o MCQ) leads to substantial drops. These results highlight the critical role of rule-based rewards in GRPO for multimodal reasoning tasks.

**Hyperparameters** To search for better hyperparameters, we experiment with different **learning rates** (LR) and **KL divergence** settings on Qwen2.5VL-3B. We start with a basic setting where LR anneals to zero following a cosine scheduler with no KL constraint. Results are shown in Table 6. LR1 uses a minimum learning rate of $8e^{-7}$ with warmup ratio 10%, whereas LR2 uses a minimum learning rate of $5e^{-7}$ with warmup ratio 3%. Since LR2 performs slightly better than LR1, we compare two KL settings on top of LR2. KL1 uses an initial KL of $1e^{-2}$ and a target KL of $5e^{-3}$, whereas KL2 uses an initial KL coefficient of $1e^{-3}$ and a target KL of $5e^{-4}$. We find that introducing KL constraints significantly improves the performance on MathVerse and DynaMath by 1.1% and 3.2%, respectively, and that using a smaller KL can encourage the model to evolve.

| Settings | MVs | DM | LV |
|---|---|---|---|
| Basic | 31.7 | 15.0 | 38.5 |
| *Learning Rate* | | | |
| + LR1 | 33.0 | 16.0 | 38.1 |
| + LR2 | 33.5 | 15.6 | 38.3 |
| *KL Coef.* | | | |
| + KL1 | 34.4 | 18.8 | 37.8 |
| + KL2 | **35.8** | 18.6 | **39.2** |

Table 6: Ablation on *LR* and *KL Coef.* on MVs: MathVerse, DM: DynaMath and LV: LogicVista.

### 4.5 Case Study

We provide an example showcasing the improvement of VLAA-Thinker over the original model in Appendix C.4. Qwen2.5VL-7B generates lengthy response with wrong reasoning traces. Although it outputs some self-reflective patterns like "re-evaluate", the final answer remains wrong. On the other hand, VLAA-Thinker-Qwen2.5VL-7B is able to reason on the right track, with only a minor mistake near the end of its thinking process. Nevertheless, the high-level idea and reasoning process is overall correct, demonstrating strong capability of solving complex reasoning tasks.

## 5 Related Work

**Vision-Language Reasoning Models.** Recent advances in vision-language (VL) reasoning models build on the success of text-only reasoning systems like OpenAI's o1 (Jaech et al., 2024) and DeepSeek-R1 (Guo et al., 2025). Earlier VL methods, such as few-shot prompting and chain-of-thought (CoT), offered limited visual reasoning (Brown et al., 2020; Wei et al., 2022). Recently, LLaVA-CoT (Xu et al., 2024) adopts an SFT approach a 4-step structured outputs to enhance model's reasoning, yet lacking flexibility due to its rigid output format. More recently, newer models incorporate more natural reasoning traces and reinforcement learning. VLM-R1 (Shen et al., 2025) and R1-V (Chen et al., 2025a) align multimodal LLMs using step-by-step reasoning and policy optimization. VisualThinker-R1-Zero (Zhou et al., 2025) goes further by training a 2B model via pure RL from scratch, achieving emergent inner reasoning. LMM-R1 (Peng et al., 2025) transfers CoT skills from language to vision through staged RL. Vision-R1 (Huang et al., 2025) combines reasoning trace supervision and RL with correctness and format rewards to train a strong 7B VL reasoner. Different from these concurrent works, we propose a high-quality multimodal reasoning dataset with R1-like reasoning traces for both SFT and RL, and provide a comprehensive study on training paradigms.

**Reward Modeling in Reinforcement Learning.** Reward design plays a central role in reasoning-oriented RL. While model-based rewards offer flexibility (Kwon et al., 2023; Wang et al., 2024a; Gao et al., 2024), they are prone to reward hacking (Eisenstein et al., 2023; Chen et al., 2024b; Fu et al., 2025), making them risky for reasoning tasks. Recent VL models prefer binary correctness rewards (Huang et al., 2025; Zhou et al., 2025) for math or QA tasks, directly reinforcing accurate outputs. Others apply rule-based rewards, enforcing structured formats or logic chains (Liu et al., 2025; Deng et al., 2025a). While recent studies deploy strong reward models for enhancing LVLM reasoning, they are grounded by specific domains or simpler tasks (Muhtar et al., 2025; Tu et al., 2025). GRPO-style methods use relative ranking within output batches to guide optimization without value critics (Shao et al., 2024; Guo et al., 2025). Our Mix Reward objective combines the model-based and rule-based reward in four complex rewarding scenarios, yielding better performance than existing approaches.

## 6 Conclusion and Limitation

This work provides a comparative analysis on the effectiveness of leveraging SFT or RL (more specifically, GRPO) to build LVLM with strong reasoning ability. We show by extensive experiments that distilling reasoning data and performing SFT is a deficient way to transfer reasoning ability across modalities. We then extend our dataset to GRPO training with a proposed mixed reward objective, which yields substantial improvement over the baseline models. We present several findings regarding combining SFT and GRPO and the correlation between reward, respond length, and final performance. These results indicate that reasoning is a native emerging ability acquired from RL, rather than SFT, which merely equips the model with "pseudo-reasoning" ability.

The major limitation of this work lies in that we only conduct experiment on Qwen2/2.5VL series, which may limit the generalization of conclusion. Also, we only experiment with one RL algorithm GRPO, whose behaviour might be different from PPO (Schulman et al., 2017) and RLOO (Ahmadian et al., 2024). Our future works should focus on expanding the generalizability on different models, as well as comparing behaviours of difference RL algorithms.

# Acknowledgement

We thank the Microsoft Accelerate Foundation Models Research Program for supporting our computing needs. Hardy Chen and Xinya Du are supported in part by the National Science Foundation CAREER Grant IIS-2340435.

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

# A    Data Generation

## A.1    Prompt

We show the prompts for captioning (Figure 8), R1 answer distillation (Figure 9), rewriting (Figure 10) and verification (Figure 11).

---

**Prompt for Captioning**

### You are a vision-language model generating a highly detailed caption of an image.
### Summarize the environment or setting (indoor/outdoor, surroundings).
### Describe visible objects, people, or structures (colors, shapes, textures, positions).
### Transcribe all text verbatim. For equations, use LaTeX when appropriate but do not solve or interpret them.
### If structured data (tables, charts) appears, use Markdown formatting for clarity.
### Include labels, annotations, brand names, or logos, if any, otherwise don't mention them.
### Note any visible expressions or emotional tone factually, without speculation.
### Maintain a logical order: from overall context to finer details.
### Provide only the caption without extra context or commentary.
### Be unbiased and faithful in your description, using natural language and Markdown only where relevant.

---

Figure 8: Prompt for captioning with GPT-4-Turbo.

---

**Prompt for Distillation**

You have advanced visual perception abilities and can directly analyze images as if you are looking at them. You will be provided with detailed visual descriptions, but you should interpret them as if they represent your actual visual understanding rather than text-based captions.

Answer questions as if you are visually perceiving the scene, not reading a caption. Provide natural and confident responses about objects, relationships, and numerical or spatial reasoning. Use a descriptive, visually grounded tone, avoiding mention of text.

Never mention that you are reading text or captions. Infer spatial relationships, numerical properties, and logical conclusions based on the perceived "image." If information is unclear, respond naturally as if there are visual limitations (e.g., 'It appears that...').

Caption:
{caption}

Question:
{question}

---

Figure 9: Prompt for distillation with Deepseek-R1.

## A.2    Aha-Moment Filtering

We use the following list of keywords to identify aha moments: `wait`, `again`, `double-check`, `hmm`, `mistake`, `alternatively`, `check`, `i should confirm`. All answers are matched with the logic: `has_aha = any([aha in text.lower() for aha in ahas])`.

To validate our idea of using "aha-moment" as a proxy of sample difficulty, we first measure the sample difficulty by repetitively sampling from Qwen-2.5-VL-32B. For each sample, we adopt the prompt in Figure 12 to generate 16 responses with `temperature` 0.6, `top_p` 0.95, `max_tokens` 1024. Thus, we obtain a pass rate for each sample. We then compute the averaged pass rate (accuracy) for SFT and GRPO split, which yield 49.9% and 45.9%. The results indicate that the GRPO split is indeed harder than the SFT split, solidifying our proposal of using "aha-moment" as a proxy of sample difficulty.

---

**Prompt for Rewriting**

You will receive a snippet of text that references a "description" or "caption" of an image. Your task is to produce a **nearly identical** version of that text with **minimal** changes, focusing on the following:

1. **Replace references to "description", "caption" and "rationale"** with wording that references **"the image."**
- For example, "The description says..." could become "The image shows..."
- "The caption suggests..." could become "The image suggests..."
- "Based on the rationale..." could become "Based on the image..."
- Make sure the replacement sounds natural but does **not** otherwise change the meaning.

2. **Preserve all line breaks, punctuation, and spacing** as much as possible, and make **no additional edits** outside of these replacements.

3. You should only output the rewritten content.

———

Here is the input:
{input}

---

Figure 10: Prompt for answer rewriting with GPT-4-Turbo.

---

**Prompt for Verification**

You are a fair evaluator.
You will be given a groundtruth and an answer from a model.
If the answer aligns with the groundtruth, output "Yes". Otherwise, output "No".
Your output should only be "Yes" or "No".

groundtruth:
{gold}

answer:
{pred}

---

Figure 11: Prompt for verification with GPT-3.5-Turbo.

## A.3 Sample Demonstration for `VLAA-Thinking-SFT-126K`

We show several examples from `VLAA-Thinking-SFT-126K` in Figure 16, Figure 17, Figure 18, Figure 19 and Figure 20.

## A.4 Quality Evaluation on SFT Split

**GPT-4o rewriting quality.** During the development stage, we eyeballed around 100 samples to optimize the prompts for rewriting. We also compute the Levenshtein distance between input and output texts, resulting in an averaged normalized similarity of 0.91. Both manual inspection and quantitative results indicate that the modification is minimal and proves that using gpt-4o for rewriting is a valid approach.

**Final data quality.** We evaluate our SFT data quality using GPT-5 on 100 randomly sampled examples (uniformly sampled from each data source) across 7 error dimensions with prompt shown in Figure 13. We find that the overall quality is 7.98 out of 10, and a rating of 8.78/10 with 84.4% high quality for mathematical samples. The low error rates demonstrate that our caption+QA approach for the text-only R1 effectively transfers visual information. The performance gap between SFT and RL stems from fundamental differences in learning paradigms: SFT teaches format imitation while RL enables genuine reasoning through exploration and feedback.

---

**Prompt for repetitive sampling with Qwen-2.5-VL-32B**

Solve the following problem by first giving a brief step-by-step analysis and then your answer. The answer in your response should be of the form ###Answer:
{image}
{question}

---

Figure 12: Prompt for repetitive sampling with Qwen-2.5-VL-32B.

---

**Prompt for Verification**

You are an expert evaluator tasked with assessing the quality of reasoning traces in visual question answering tasks. Your job is to identify specific types of errors in the reasoning process.

**TASK**: Evaluate the following reasoning trace for a visual question answering problem.
**QUESTION**: {question}
**REASONING TRACE**: {reasoning_trace}
**EVALUATION CRITERIA**: Please evaluate the reasoning trace and count the number of errors in each of the following categories:
1. **FACTUAL_ERRORS**: Incorrect statements about the image content, object properties, or visual details
2. **LOGICAL_ERRORS**: Flawed logical reasoning, incorrect mathematical operations, or invalid deductions
3. **INCOMPLETE_REASONING**: Missing steps, incomplete analysis, or failure to address all aspects of the question
4. **INCONSISTENCY_ERRORS**: Contradictory statements within the reasoning trace
5. **HALLUCINATION_ERRORS**: Describing objects, properties, or details that are not present in the image
6. **PROCESSING_ERRORS**: Errors in understanding the question, misinterpreting instructions, or wrong problem formulation
7. **VERIFICATION_ERRORS**: Failure to double-check calculations, assumptions, or conclusions

**INSTRUCTIONS**:
- Carefully read through the entire reasoning trace
- Identify specific instances of each error type
- Count the total number of errors in each category
- Be precise and conservative in your evaluation
- If no errors of a particular type are found, return 0 for that category
- Focus on the reasoning process, not just the final answer

**OUTPUT FORMAT**:
Return your evaluation as a JSON object with the following structure:
{{
"factual_errors": <number>,
"logical_errors": <number>,
"incomplete_reasoning": <number>,
"inconsistency_errors": <number>,
"hallucination_errors": <number>,
"processing_errors": <number>,
"verification_errors": <number>,
"overall_quality_score": <score from 1-10>,
"explanation": "<brief explanation of the main issues found>"
}}

**IMPORTANT**:
- Return ONLY the JSON object, no additional text
- Ensure the JSON is valid and properly formatted
- The overall_quality_score should be from 1 (very poor) to 10 (excellent)
- Be objective and fair in your assessment

---

Figure 13: Prompt for verification with GPT-3.5-Turbo.

# B   Details of SFT Experiments

## B.1   Training

To enhance the instruction following ability, we append task-specific instructions (*i.e.*, MCQ, short answer) to questions. The system prompt shown in Figure 14 is used. We use a global batch size of 128. Models are trained for 190 steps on 25K samples and 985 steps on 126K samples. All experiments are run on 8*H100 GPUs.

Interestingly, we observe loss spikes for 25K SFT training on Qwen2VL-7B which causes model collapse. Therefore, we run the settings for multiple times until we obtain a normal loss curve, and use that checkpoint for evaluation.

> You are VL-Thinking🤔, a helpful assistant with excellent reasoning ability. A user asks you a question, and you should try to solve it. You should first think about the reasoning process in the mind and then provides the user with the answer. The reasoning process and answer are enclosed within <think> </think> and <answer> </answer> tags, respectively, i.e., <think> reasoning process here </think> <answer> answer here </answer>.

Figure 14: System Prompt used for training and evaluation.

## B.2   Evaluation

We adopt VLMEvalKit (Duan et al., 2024) for all evaluation experiments. We set `use_custom_prompt` to `False` following the settings of most models in the toolkit. For higher efficiency, we set `max_pixels` to 256*32*32, and `max_new_tokens` to 800. We also set system prompt as the one we used for training for a consistent training-test behavior. The other hyperparameters are default to the original toolkit.

We specify the split of datasets and metrics reported:

1. MathVista: The Test Mini split of MathVista dataset; overall accuracy.

2. MathVision: The Full test set of MathVision; overall accuracy.

3. MathVerse: The Test Mini split of MathVerse; accuracy of "Vision Only" .

4. DynaMath: The Full test set of DynaMath; overall accuracy.

5. WeMath: The Test Mini split of WeMath; "Score (Strict)".

6. LogicVista: The Full test set of LogicVista; overall accuracy.

# C   Details of GRPO Experiments

## C.1   Training

We adapt our code from OpenRLHF framework (Hu et al., 2024). To suit for our need of deploying a reward model on the same machine, we offload the reward model to CPU and only move it to GPU when performing rollouts and scoring. This design saves valuable GPU memory which accelerate the training process.

We also perform dataset-specific inspection and find some issues for several datasets. For example, although ArxivQA contains only MCQ, the answer format includes "A", "A)", "(a)", *etc.* And in the synthesis subset of Math PUMA, we find that some solutions only contain the value of solved unknown variables when

the questions ask to output the entire function expression. We fix these issues by rule-based filtering and GPT-assisted rewriting, aiming to improve the quality of the VL-Thinking dataset.

## C.2 Evaluation

We evaluate our models with an identical setting described in Appendix B.2.

## C.3 Generalization Beyond Math Benchmarks

We evaluate model performance on two splits of VCR (Zhang et al., 2025), which is a complex reasoning task on restoring occluded texts. Results are presented in Table 7, which shows that on both 3B and 7B models, GRPO is more effective than SFT+GRPO. Even though both settings show degradation on VCR-En-Easy for 7B models over the baseline, GRPO shows less degradation than SFT+GRPO.

| Model | VCR-EN-Easy | VCR-EN-Hard | Avg |
|---|---|---|---|
| *3B Models* | | | |
| Qwen2.5-VL-3B-Inst | 0.69 | 0.03 | 0.36 |
| SFT+GRPO | 0.23 | 0.12 | 0.18 |
| **GRPO (ours)** | **0.73** | **0.38** | **0.56** |
| *7B Models* | | | |
| Qwen2.5-VL-7B-Inst | **0.85** | 0.21 | 0.53 |
| SFT+GRPO | 0.58 | 0.38 | 0.48 |
| **GRPO (ours)** | 0.66 | **0.49** | **0.58** |

Table 7: Performance comparison on VCR-EN-Easy and VCR-EN-Hard for 3B and 7B models.

## C.4 Case Study

We present a case demonstrating the improvement of VLAA-Thinker-Qwen2.5VL-7B over its backbone in Figure 15.

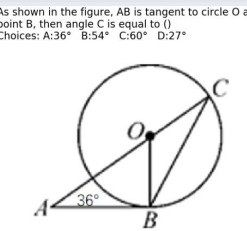

As shown in the figure, AB is tangent to circle O at point B, then angle C is equal to ()
Choices: A:36° B:54° C:60° D:27°

**Question in Image**

As shown in the figure, if angle A = 36.0, then angle C is equal to ()
Choices: A:36° B:54° C:60° D:27°

**Qwen2.5VL-7B**

To solve the problem, we need to use some properties of circles and tangents.

1. **Identify the given information:**
   - AB is tangent to circle O at point B.
   - The angle $\angle AOB = 36°$.
2. **Understand the properties:**
   - Since $AB$ is tangent to the circle at $B$, the radius $OB$ is perpendicular to the tangent $AB$. Therefore, $\angle OBA = 90$.
   - In triangle $OAB$, the sum of the angles is 180°. So, we can find $\angle AOB$:
$$\angle AOB + \angle OBA + \angle OAB = 180°$$
$$36° + 90° + \angle OAB = 180°$$
$$\angle OAB = 54°$$
3. **Determine the measure of $\angle C$**
   - Since $\angle OAB$ is an inscribed angle that intercepts the arc $BC$, the measure of the central angle $\angle BOC$ that intercepts the same arc $BC$ is twice the measure of the inscribed angle $\angle OAB$:
$$\angle BOC = 2 \times \angle OAB = 2 \times 54° = 108°$$
   - Now, in triangle $OBC$, the sum of the angles is 180°. So, we can find $\angle C$:
$$\angle OBC + \angle OCB + \angle BOC = 180°$$
$$90° + \angle OCB + 108° = 180°$$
$$\angle OCB = 180° - 198° = 18°$$
However, since $\angle OCB$ is not directly given as an option, we need to re-evaluate the problem with the correct interpretation of the angle $\angle C$ which is actually the angle subtended by the arc BC at the circumference, which is half the central angle:
$$\angle C = \frac{1}{2} \times \angle BOC = \frac{1}{2} \times 108° = 54°$$
Therefore, the measure of $\angle C$ is 54°.
The answer is: B

**VLAA-Thinker-Qwen2.5VL-7B**

*<think>*
To solve this problem, we need to use the properties of tangents and circles. Since AB is tangent to circle O at point B, the radius OB is perpendicular to the tangent AB at point B. This means that angle OBA is 90°.

that angle A is 36°, we can find angle AOB by subtracting angle A from 180° (since the sum of angles in triangle AOB must be 180°):
$$\angle AOB = 180° - 36° - 90° = 54°$$
Since angle AOB is a central angle subtending arc BC, and angle C is an angle subtending the same arc BC, the measure of angle C is half the measure of angle AOB:
$$\angle C = \frac{1}{2} \times 54° = 27°$$
Therefore, the correct answer is D: 27°.
*</think>*
*<answer>*
D: 27°
*</answer>*

Figure 15: A case from MathVerse testmini (sample index 20). Markdowns are rendered for illustration purpose. Wrong reasoning paths are colored red.

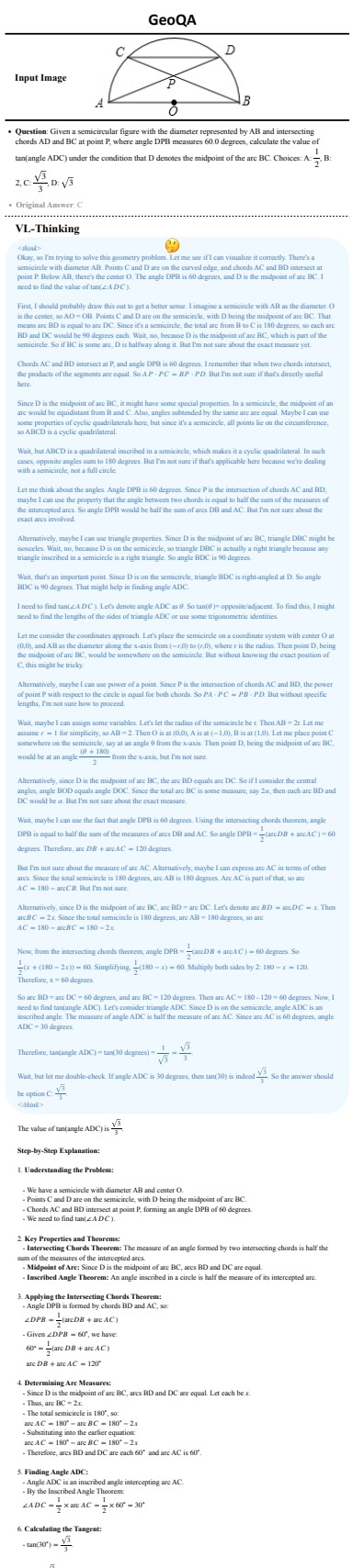

Figure 16: A VL-Thinking sample from GeoQA170K.

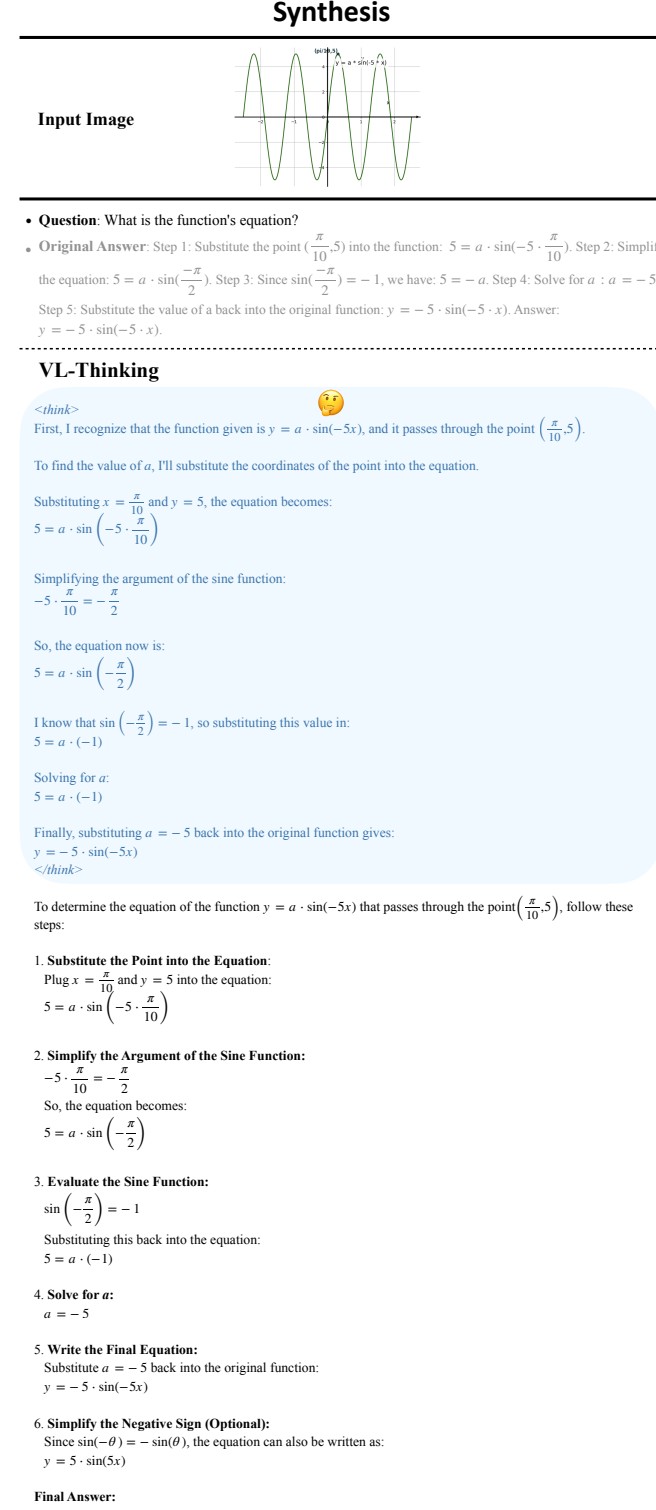

Figure 17: A VL-Thinking sample from Math PUMA (subset Synthesis).

# CLEVR-MATH

**Input Image**

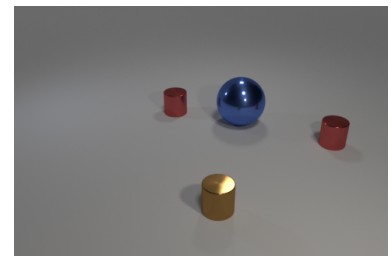

- **Question**: Subtract all balls. How many objects are left?

- **Original Answer**: 3

## VL-Thinking

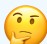

*<think>* Okay, let's see. The user is asking to subtract all balls and determine how many objects are left. First, I need to figure out which objects are considered "balls" in the image. The image shows a large blue metal sphere. A sphere is a three-dimensional circle, which is essentially a ball shape. So the blue metal sphere is definitely a ball.

Next, the surrounding objects are two small red metal cylinders and a single small brown metal cylinder in the foreground. Cylinders are not balls; they have a different shape—circular bases and straight sides. So those would not be subtracted. The question says to subtract all balls, so only the sphere (ball) needs to be removed.

Originally, there's the sphere (1 object), two red cylinders (2 objects), and one brown cylinder (1 object). Total objects before subtraction: $1 + 2 + 1 = 4$. Subtracting the one ball leaves 3 objects: the two red cylinders and the brown cylinder.

Wait, but I should make sure there are no other balls mentioned. The image only refers to the sphere as a ball-like object. The cylinders are clearly not balls. So subtracting the sphere leaves three cylinders. The answer should be 3.
*</think>*

In the scene, there is one large blue sphere (a ball-shaped object) centrally positioned. The remaining objects consist of two small red cylinders flanking it symmetrically and a single brown cylinder in the foreground. Removing the sphere leaves only the three cylindrical objects visible.

**Answer:** There are 3 objects remaining.

Figure 18: A VL-Thinking sample from CLEVR-Math.

# ArxivQA

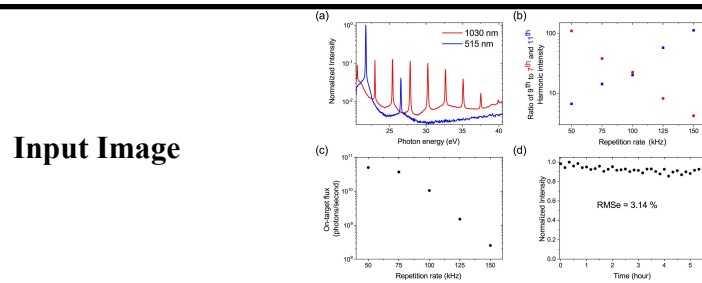

**Input Image**

- **Question**: In figure (b), how does the repetition rate affect the ratio of 9th to 11th harmonic intensity for the 1030 nm wavelength? Choose from the following options A) The ratio increases as the repetition rate increases. B) The ratio decreases as the repetition rate increases. C) The ratio remains constant as the repetition rate increases. D) The ratio fluctuates unpredictably with changes in repetition rate.

- **Original Answer**: B

---

## VL-Thinking

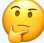

*<think>*

Okay, so I'm looking at figure (b) here, and I need to figure out how the repetition rate affects the ratio of 9th to 11th harmonic intensity for the 1030 nm wavelength. Let me start by understanding what's in the image.

First, the image is a scatter plot with the x-axis being the repetition rate in kHz and the y-axis being the ratio of 9th to 7th and 11th harmonic intensity on a logarithmic scale. There are two sets of data points: red dots and blue squares. The red dots are probably representing the 1030 nm wavelength since in panel (a), the red curve was labeled as 1030 nm. The blue squares would then correspond to the 515 nm wavelength.

The question specifically asks about the 1030 nm wavelength, so I'll focus on the red dots. Looking at the scatter plot, as the repetition rate increases from left to right on the x-axis, I need to see what happens to the y-axis values for the red dots.

From what I can see, as the repetition rate goes up, the red dots trend downward. That means the ratio of 9th to 11th harmonic intensity is decreasing. There's no indication of an increase or a constant ratio; instead, it's a clear downward trend. The red dots don't seem to fluctuate unpredictably either; they follow a decreasing pattern as repetition rate increases.

So, putting it all together, the ratio decreases as the repetition rate increases for the 1030 nm wavelength. That would correspond to option B.

*</think>*

The image shows that the ratio of 9th to 11th harmonic intensity for the 1030 nm wavelength decreases as the repetition rate increases.

**Answer: B) The ratio decreases as the repetition rate increases.**

Figure 19: A VL-Thinking sample from ArxivQA.

# ALLaVA-LAION

**Input Image**

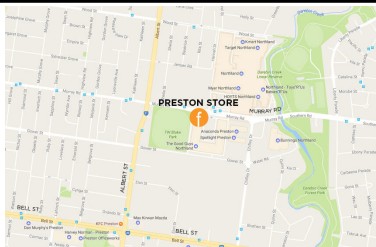

- **Question**: Considering the proximity of Preston Store to other landmarks, what could be its significance in the local area?

- **Original Answer**: The Preston Store's significance in the local area, as suggested by the map, could be quite central given its prominent placement and the fact that it has a dedicated Facebook page marker. It appears to be situated near major roads and commercial entities like KFC, Harvey Norman, and Dan Murphy's, which indicates it is likely a well-known establishment, possibly a retail store or a social venue, that serves as a local landmark or gathering place. The specific type of store cannot be determined from the map alone, but its central location among various retailers and close to a park suggests it may cater to a wide range of customers and possibly holds a key place in the local community.

### VL-Thinking

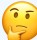

*<think>*

Okay, let's see. The user is asking about the significance of Preston Store in the local area based on its proximity to other landmarks. First, I need to visualize the map shown. The Preston Store is labeled prominently with a red marker and a Facebook 'f' logo, so it's probably a key location.

Looking at the nearby landmarks, there's KFC, Harvey Norman, Dan Murphy's, and Max Kirwan Mazda. Those are all big-name retailers and services. Then there's Target, Myer Northland, and Anaconda Preston. These are major stores, which suggests the area is a commercial hub. The store being near these places means it's likely in a busy shopping district, making it a central spot for locals.

Also, there's TW Blake Park and Darebin Creek Forest Park nearby. Green areas mean that Preston Store might be a place people visit before or after outdoor activities. Maybe it's a convenience store or a community spot where people gather. The mix of retail, dining, and parks indicates it's a multifunctional area, so Preston Store could serve both daily shopping needs and act as a meeting point. The Facebook marker suggests they have an online presence, maybe promoting local events or services, enhancing community engagement. Since it's near major roads like Bell St and Murray Rd, accessibility is good, increasing its importance as a central landmark.

*</think>*

Preston Store appears centrally positioned within a bustling commercial and recreational hub, making it a likely focal point for local activity. Its prominent placement near major retailers like Target, Myer Northland, and Harvey Norman suggests it could serve as a practical anchor for shoppers—perhaps offering essentials or services that complement these larger stores. The proximity to fast food (KFC) and leisure destinations (Darebin Creek Forest Park, TW Blake Park) positions it as a potential pitstop for families or individuals transitioning between errands, dining, and outdoor activities. The red Facebook marker implies community engagement, possibly acting as an information node or local business with social media visibility. Its accessibility via intersecting roads like Bell St and Murray Rd further underscores its role as a convenient landmark in the area's daily flow.

Figure 20: A VL-Thinking sample from ALLaVA-LAION.

