# OpenReview forum: "SFT or RL? An Early Investigation into Training R1-Like Reasoning Large Vision-Language Models"
_TMLR — Accepted by TMLR_

### Review · Reviewer_7XcF · 2025-07-22

**Summary Of Contributions:**

The paper investigates the two stage procedure of LVLM training for reasoning tasks: SFT followed by RL. The paper does that by systematically introducing a multimodal reasoning dataset “VL-Thinking” dataset that leverages GPT-4o, DeepSeek-R1, GPT-3.5 at different steps of dataset curation. The dataset is split into spfecific SFT and RL sets such that complex samples are used for SFT to avoid imitation learning. The authors conduct experiments comparing SFT, RL and combination of both using Qwen2VL, Qwen2.5VL on various visual math reasoning benchmarks. The paper also introduces mixed reward module consisting of rule based rewards, open ended reward, implicit format reward. The paper presents key findings such as SFT before GRPO leads to worse performance than GRPO alone, and SFT constrains exploration paths for reasoning models.

**Audience:**

Yes

**Claims And Evidence:**

Yes

**Requested Changes:**

Apart from addressing the weakness above. I would suggest the authors to address the following points:

- Majority of the benchmarks are maths reasoning benchmarks, does the findings from this paper hold for common sense reasoning etc,
- Explain the VL-Thinking dataset curation in more detail, especially the key details related to the DeepSeek-R1 based reasoning traces and SFT/RL split based on “aha moment”.
- I would suggest the authors to also put limitations and future work for the current study.

**Strengths And Weaknesses:**

Strengths:
- The paper addresses a fundamental question in LVLM training of whether of SFT or RL leads to reasoning capabilities in Large models.
- The paper introduces a new multimodal reasoning dataset namely VL-Thinking. This dataset will be of much value for future research in multimodal reasoning.
- The paper proposed mixed reward module for GRPO training, that combines rule based rewards, open ended rewards, and implicit format reward.
- The paper shows response length and initial reward values do no correlate with reasoning capability in RL trained models.

Weaknesses:
- The VL-Thinking dataset reasoning traces are distilled from DeepSeek R1. SFT can imitate suboptimal reasoning patterns from DeepSeekR1. A thorough analysis on this is crucial to make claims in this paper more accurate.
- The paper does not show any comparisons with SOTA RL based LVLm reasoning methods.
- While the empirical findings in this paper show that SFT is detrimental to reasoning capability of the LVLM but the authors do not explore why it might be the case. Can it be related to catastrophic forgetting and if yes what are potential methods to avoid this issue?
- The split between SFT and RL is crucial and must be explained in more detail.
- Its not clear from the paper, whether the authors propose for SFT related degradation. Do the authors suggest to get away from SFT to RL completely or can the authors suggest any strategy to avoid the detrimental effect of SFT?

---

> ### Author Response · Authors · 2025-09-11
> **Response (1/2)**
>
> > 1. Distillation from text-only R1 may not be optimal.
>
> Thanks for raising the question. Please refer to General Response 2 where we evaluate the quality of R1-generated samples.
>
> > 2. Comparison with other RL-based models.
>
> Both Taichu-VLR-3B and VLM-R1-Math-0305 in Tab4 are RL-based LVLM baselines for 3B models. In addition, we add the results of R1-OneVision [1] as one of our 7B baselines. Results in the table below show that our model outperforms R1-OneVision by a large margin, suggesting the effectiveness of our approach.
>
>
> | Model  |   MathVista |   MathVision |   MathVerse |   DynaMath |   WeMath |   LogicVista |  Avg. |
> |:--------------|------------:|-------------:|------------:|-----------:|---------:|-------------:|--------:|
> | R1-OneVision  |        65.4 |        22.73 |       41.24 |      14.97 |    20.76 |        42.06 | 34.53   |
> | VL-Thinker-7B (ours) |       **68.0**|        **26.40** |       **48.20** |      **22.40** |    **41.50** |        **48.50** | **42.50**   |
>
>
> [1] Yang, Yi, et al. "R1-onevision: Advancing generalized multimodal reasoning through cross-modal formalization." arXiv preprint arXiv:2503.10615 (2025).
>
>
>
> > 3. Why SFT degrades performance.
>
> We provided analysis in Sec4.2 and we restate it here for clarity. We conjecture that degraded performance of SFT may stem from the **imitative** nature of its training objective, which enforces a model to imitate reasoning traces in a token-by-token fashion. On the other hand, RL is an **explorative** method that only rewards the model based on the final outcome. Since reasoning is conceivably a native emerging ability, RL-based methods would be a better fit for this scenario. We provide practical implications in question 5 below.

---

> ### Author Response · Authors · 2025-09-11
> **Response (2/2)**
>
> > 4. Dataset split clarity: The SFT/RL split is crucial; need more detailed explanation (esp. use of “aha moments”).
>
> To validate our idea of using “aha-moment” as a proxy of sample difficulty, we first measure the sample difficulty by doing repetitive sampling with Qwen-2.5-VL-32B. For each question, we adopt the following prompt
> ```
> Solve the following problem by first giving a brief step-by-step analysis and then your answer. The answer in your response should be of the form ###Answer: Answer (without quotes) where Answer is the answer to the problem.\n\n"
> ```
> to generate **16** responses with temperature 0.6, top_p 0.95, max_tokens 1024. Thus, we obtain a pass rate for each sample. We then compute the averaged pass rate for SFT and GRPO split, which yield 49.9% and 45.9%. The results indicate that the GRPO split is indeed harder than the SFT split, solidifying our proposal of using “aha-moment” as a proxy of sample difficulty.
>
>
> > 5. Practical implications: Are authors suggesting to abandon SFT entirely? Or are there strategies to avoid degradation? Clarify stance.
>
> Based on our experiment setting and results, we clarify our stance as follows. If the model is able to explore answer formats via RL, then we shall skip reasoning SFT so that the model can explore reasoning traces by itself rather than via imitating. For implementation, one should experiment with different reasoning SFT+RL strategies rather than abandoning SFT entirely.
>
>
>
>  > 6. Generalization beyond math benchmarks.
>
> To address your concern, we additionally test on VCR benchmark [1], which is a complex reasoning task on restoring occluded texts and is supported in VLMEvalKit [2].
>
> | Model                  | VCR-EN-Easy | VCR-EN-Hard | Avg  |
> |-------------------------|-------------|-------------|------|
> | Qwen2.5-VL-3B-Instruct | 0.69        | 0.03        | 0.36 |
> | SFT+GRPO (3B)          | 0.23        | 0.12        | 0.18 |
> | GRPO (3B)              | **0.73**    | **0.38**    | **0.56** |
>
> | Model                  | VCR-EN-Easy | VCR-EN-Hard | Avg  |
> |-------------------------|-------------|-------------|------|
> | Qwen2.5-VL-7B-Instruct | **0.85**    | 0.21        | 0.53 |
> | SFT+GRPO (7B)          | 0.58        | 0.38        | 0.48 |
> | GRPO (7B)              | 0.66        | **0.49**    | **0.58** |
>
> Results on both 3B and 7B models show that GRPO is more effective than SFT+GRPO. Even though both settings show degradation on VCR-En-Easy for 7B models over the baseline, GRPO shows less degradation than SFT+GRPO.
>
>
> [1] Zhang, Tianyu, et al. "VCR: A Task for Pixel-Level Complex Reasoning in Vision Language Models via Restoring Occluded Text." arXiv preprint arXiv:2406.06462 (2024).
>
> [2] Duan, Haodong, et al. "Vlmevalkit: An open-source toolkit for evaluating large multi-modality models." Proceedings of the 32nd ACM international conference on multimedia. 2024.
>
>
>
> > 7. Add limitations and future work.
>
> Thanks for your suggestion. As stated in General Response 1, the major limitation of this work lies in that we only did experiment on Qwen2/2.5VL series, which may limit the generalization of conclusion. Also, we only experiment with one RL algorithm GRPO, whose behaviour might be different from PPO [1] and RLOO [2]. Our future works should focus on expanding the generalizability on different models, as well as comparing behaviours of difference RL algorithms.
>
> [1] Schulman, John, et al. "Proximal policy optimization algorithms." arXiv preprint arXiv:1707.06347 (2017).
>
> [2] Ahmadian, Arash, et al. "Back to basics: Revisiting reinforce style optimization for learning from human feedback in llms." arXiv preprint arXiv:2402.14740 (2024).

---

### Review · Reviewer_6XCA · 2025-07-24

**Summary Of Contributions:**

This study aims to answer two central questions: "What are the distinct effects of SFT and RL in multimodal reasoning?" and "Is this two-stage paradigm truly necessary for reasoning in Large Vision Language Models (LVLMs)?" To this end, the authors build reasoning models for vision-language models (VLMs) using both supervised fine-tuning (SFT) and reinforcement learning (RL). They introduce a new multimodal dataset, VL-Thinking, which is constructed through a six-stage pipeline including captioning, reasoning distillation, answer rewrite, and verification. The dataset contains more challenging examples for RL than for SFT. The authors claim that while SFT helps models learn reasoning formats, it often locks aligned models into imitative, rigid reasoning modes that impede further learning. In contrast, they report that reinforcement learning with a mixed reward module and Group Relative Policy Optimization (GRPO) enables more natural and flexible reasoning, achieving higher performance. Their model, VL-Thinker, based on Qwen2.5VL 3B, achieves state-of-the-art performance across six popular visual math reasoning benchmarks among 4B-scale LVLMs.

**Audience:**

Yes

**Broader Impact Concerns:**

None.

**Claims And Evidence:**

No

**Requested Changes:**

Please reconsider the main claims of this work (see the weaknesses of this paper).

**Strengths And Weaknesses:**

## Strengths

+ The proposed VL-Thinker model outperforms similarly sized models on the Open LMM Reasoning Leaderboard.
+ The RL approach adopted in this work (i.e., the use of GRPO and the reward design described in Section 4.1) is reasonable.
+ The paper is well-written and easy to follow.

## Weaknesses

While the paper presents interesting insights, its central claims suffer from overgeneralization and questionable assumptions. In its current form, the work may not be suitable for journal publication.

First, the central hypothesis that "SFT can significantly undermine subsequent RL" is problematic. According to Section 2, the datasets used for SFT and RL are disjoint, and the RL dataset was explicitly designed to be more challenging. If the training materials for the two differ substantially, then comparing SFT and RL is not fair. Favoring the more difficult examples for RL puts SFT at an unfair disadvantage. Furthermore, Table 3 shows that applying GRPO to Qwen2VL-7B-Instruct yields better performance than applying it to Qwen2VL-7B-Base. If the former includes SFT in its construction of Qwen2-VL-7B-Instruct, this result contradicts the claim that SFT impedes RL. It is thus plausible that the issue lies not in SFT per se, but rather in the specific SFT dataset used in this study. While I see that it is difficult to ensure fair conditions when comparing SFT and RL, the decision to base the entire paper on such a hypothesis may have been inappropriate.

Second, the claim that "reasoning is a native emerging ability that is more likely to be developed through RL, not SFT" seems overly broad. Since the experiments are conducted solely on vision-language models, it would be more appropriate to support this claim with additional experiments in the text-only setting. Without this, generalizing the conclusion to reasoning models at large is potentially misleading.

The explanation of the VL-Thinking dataset (Section 2) states that DeepSeek-R1 was used for Reasoning Answer Distillation and GPT-3.5 for Answer and Rewriting. However, the paper does not justify the choice of these LLMs for reasoning data synthesis, nor does it evaluate the quality of the resulting dataset.

The paper also states in Section 2 that "we propose using the presence of self-reflective cues (i.e., the 'aha moments') in the distilled answers as an indicator of a sample's difficulty level." If this is indeed a proposal, empirical validation is needed, but none is provided.

In Section 3, the authors write that "Our finding suggests that self-reflection thinking ('aha moments') from the SFT process is overloaded." However, if the SFT dataset was constructed from relatively simple examples with few self-reflective cues, it is unsurprising that the model trained on it would produce limited or unhelpful self-reflection outputs.

Finally, while achieving top scores on the Open LMM Reasoning Leaderboard is a contribution, the RL method employed (GRPO with outcome-based reward) does not appear to offer significant novelty compared to the work of DeepSeek-R1.

---

> ### Author Response · Authors · 2025-09-11
> **Response (1/2)**
>
> > 1. Central hypothesis (“SFT undermines RL”) is problematic: Because SFT and RL datasets are disjoint, with RL having harder data. This comparison may be unfair and need to clarify or reframe.
>
> We have shown in Tab2 that SFTing on 55K samples with aha moments (`w/ aha-55K`) yields similar degree of degradation to SFTing on 25K samples without aha moment (`w/ 25K`). Despite `w/ aha-55K` being a harder split than `w/ 25K`, it does not gain any advantage over the latter (21.3 VS 21.6). Therefore, we use 25K samples as the default dataset for SFT in the subsequent experiments, and this practice can ensure that SFT gains a reasonable performance that promotes fairness in its comparison with GRPO.
>
>
> > 2. Contradiction with results in Table 3: GRPO on Qwen2VL-7B-Instruct outperforms GRPO on Base, despite SFT being part of Instruct. This undermines the claim that SFT always harms RL. Need to reconcile.
>
> Thanks for pointing out the conflict between results in Tab3 and our conclusion. After refining our claim, the conflicts should be addressed.
> - **GRPO on Inst is better than Base (line 1 vs line 3 in Tab3)**
> We conjecture that this is because GRPO requires certain formats specified in the system prompt, and Inst model can find the correct format via exploration more quickly than Base model, thus the former would have more steps to learn reasoning given the same number of training steps.
>
> - **GRPO on reasoning-SFTed Inst is worse than reasoning-SFTed Base (line 2 vs line 4 in Tab3)**:
> This result supports our claim that reasoning-SFT before GRPO would hurt performance on math reasoning benchmarks.
>
>
> > 3. Overgeneralization of “reasoning emerges only via RL”: Currently applied beyond VL to all reasoning models. Should limit scope to VL or provide evidence from text-only setting.
>
> Thanks for your suggestion! We refine our conclusion in General Response 1 to avoid the over-claiming issue.
>
>
> > 4. Dataset synthesis choices: Justify why DeepSeek-R1 was chosen for distillation and GPT-3.5 for rewriting, and evaluate their quality.
>
> - **Why DeepSeek-R1 for distillation:**
> Deepseek-R1 was the most capable reasoning model when our work was conducted. Meanwhile, inspired by [1] which finds that replacing images with captions and feeding them to an LLM can achieve higher performance than models with multimodal inputs, we opt to prompt the strong reasoning model R1 to curate a vision-language reasoning dataset.
>
> - **Quality of distilled datasets:** Please refer to General Response 2.
>
> - **Why GPT-3.5 for rewriting:**
> This is a typo in our manuscript (step #4 in Sec4). In fact, we used **`gpt-4o`** for rewriting, and the prompt is shown in Fig10. We will fix the typo in our camera-ready version. Rewriting is a relatively simple and straightforward task. With clear instructions provided, gpt-4o should be able to output decent answers.
>
> - **Quality of rewriting using gpt-4o**:
> During the development stage, we eyeballed ~100 samples to optimize the prompts for rewriting. To provide a qualitative result, we compute the Levenshtein distance between input and output texts, resulting in **an averaged normalized similarity of 0.91**. Both manual inspection and quantitative results indicate that the modification is minimal and proves that using gpt-4o for rewriting is a valid approach.
>
>
> [1] Berrios, William, et al. "Towards language models that can see: Computer vision through the lens of natural language." arXiv preprint arXiv:2306.16410 (2023).

---

> > ### Author Response · Authors · 2025-09-11
> > **Response (2/2)**
> >
> > > 5. “Aha-moment” difficulty proxy needs empirical validation.
> >
> > To validate our idea of using “aha-moment” as a proxy of sample difficulty, we first measure the sample difficulty by repetitively sampling from Qwen-2.5-VL-32B. For each sample, we adopt the following prompt
> > ```
> > Solve the following problem by first giving a brief step-by-step analysis and then your answer. The answer in your response should be of the form ###Answer: Answer (without quotes) where Answer is the answer to the problem.\n\n"
> > ```
> > to generate **16** responses with temperature 0.6, top_p 0.95, max_tokens 1024. Thus, we obtain a pass rate for each sample. We then compute the averaged pass rate (accuracy) for SFT and GRPO split, which yield 49.9% and 45.9%. The results indicate that the GRPO split is indeed harder than the SFT split, solidifying our proposal of using “aha-moment” as a proxy of sample difficulty.
> >
> >
> >
> > > 6. SFT reasoning traces: If SFT dataset is simple and lacks reflective cues, it’s unsurprising that SFTed models show limited reflection. Need to acknowledge and analyze this.
> >
> > In Tab2, we SFT Qwen2.5VL-3B on 55K samples, all of which contain reflective cues (w/ aha-55K). Results show that it performs on par with SFTing on 25K samples *without* reflective cues (w/ 25K). As stated in the first paragraph of section 3.2, self-reflective cues learned via SFT are superficial and unreliable. Besides, we also find that the overall inference time of models trained under the *w/ aha-55K* setting is 10x longer than others. By examining model outputs, we find meaningless repetitive reflective patterns which persist until maximum generation length 4096 is reached. This is an evidence that supports 1) more reflective cues do not lead to higher performance and 2) reflective cues learned via SFT imitation harms model performance and makes inference inefficient.
> >
> >
> > > 7. Novelty of method: GRPO with outcome-based reward is not highly novel compared to DeepSeek-R1. Need to better emphasize what’s new.
> >
> > We summarize our novelties as follows:
> > - Different from R1, we find that reasoning SFT before GRPO hurts model performance in VL reasoning tasks.
> > - We design a set of 5 outcome-based verifiable rewards for VL training (see Fig4). The reward system consists of 2 types of rewards: rule-based and open-ended, while R1 paper consists of only rule-based reward, which limits the generalization to open-ended tasks. Ablation in Tab5 shows that open-ended reward is crucial for final performance (row 5 VS row 6).

---

### Review · Reviewer_M1xC · 2025-08-28

**Summary Of Contributions:**

The authors investigate the prevalent "SFT-then-RL" paradigm for training vision language models. They construct a large-scale dataset suitable for both supervised fine-tuning (SFT) and reinforcement learning (RL) using an automated data collection pipeline. Through experiments with Qwen2 and Qwen2.5-based vision language models, they find that applying RL without prior SFT achieves competitive state-of-the-art results, improving average performance on math-focused vision language benchmarks compared to similarly sized models. Their analysis further shows that SFT alone can significantly degrade model performance. Notably, on Qwen2-VL-7B and Qwen2.5VL-3B, RL without SFT consistently outperforms the conventional pipeline of SFT followed by RL.

**Audience:**

Yes

**Broader Impact Concerns:**

No concerns.

**Claims And Evidence:**

No

**Requested Changes:**

Critical points:

- Update the framing and implications: The conclusions are currently drawn solely from experiments on Qwen-based models. Some of the observed behaviors may not generalize to models built on other base LLMs, and this limitation should be explicitly acknowledged.
- Clarify model usage across experiments: The presentation of results does not always make it clear which models are evaluated in each section. Improving transparency here is essential for interpreting the findings.
- Discuss the influence of fine-tuning data: How might the fine-tuning dataset impact the benchmark results? For example, the most notable improvements occur on MathVerse and WeMath, while other benchmarks show marginal or even negative changes. What might explain this? One hypothesis could be that the model’s ability to handle synthetic inputs improves, given that these datasets primarily consist of synthetic black-and-white images. A discussion along these lines would strengthen the paper considerably.

Non-critical points
- Extending the experiments to another model that is not Qwen-based would be very insightful. However, I understand that it could go beyond the scope.
- One of the rewards is called "Rule-based reward". It appears to function more like a supervised, verifiable reward rather than a truly rule-based one (which suggests execution of explicit rules on top of model outputs). Refining this terminology would avoid confusion.

Small remarks:
- The abbreviation MCQ is never introduced.
- The color scheme from Fig. 5 could be used for Fig. 3 as well.
- Table 4, lower part, misses the best performance marked for MathVista.

**Strengths And Weaknesses:**

Strengths:
- The proposed VL-Thinking dataset represents a valuable contribution, providing a useful resource both for fine-tuning vision-language models (VLMs) and for systematically studying the effects of different fine-tuning strategies.
- The experimental results offer notable insights into model behavior under varying fine-tuning approaches.
- The paper is well-written, clearly structured, and easy to follow, which makes the contributions accessible.

Weaknesses:
- The choice of models across experiments sometimes appears inconsistent or arbitrary. For example, the first two findings in Section 3.2 rely only on Qwen2.5-VL-3B, while Section 4.2 presents results for Qwen2-VL-7B (Instruct and Base) before returning to Qwen2.5-VL-3B only. While the high computational cost of such evaluations is understandable, the presentation could be clearer about which models are evaluated for which findings.
- The findings are often generalized too broadly, as if they would necessarily hold across all VLMs during fine-tuning. However, no discussion of limitations is provided. For instance, results could vary with different training data, evaluation datasets, or base LLM architectures. Prior work (Shao et al.) already suggests that such differences can be significant.

Shao, Rulin, et al. "Spurious rewards: Rethinking training signals in rlvr." arXiv preprint arXiv:2506.10947 (2025).

---

> ### Author Response · Authors · 2025-09-11
> **Response**
>
> > 1. Limitations of generalization: Current conclusions are drawn only from Qwen-based models; must explicitly acknowledge that results may not generalize to other architectures or datasets.
>
> Thanks for pointing out the limitations of our work. We have refined our conclusion in General Response 1, and we will add a limitation section in the updated version.
>
> > 2. Model usage in sec3.2 and sec4.2 needs clarification
>
> Thanks for pointing out the issue on choices of models. In Sec3.2, we choose Qwen2.5-VL-3B because it is a capable small model which allows us to conduct ablation studies at less cost. In Sec4.2, we choose Qwen2 series because Qwen2.5 series do not have Base models. We choose Qwen2-7B because we find that GRPO on Qwen2-2B-Base and Qwen2-2B-Base+SFT yield model collapse, which prevents us from drawing conclusions on choices of backbone.
>
> > 3. Impact of fine-tuning dataset: Discuss why improvements are strong on MathVerse/WeMath but weaker or negative elsewhere.
>
> Thank you for raising this question. The stronger improvements observed on MathVerse and WeMath compared to other vision-language reasoning benchmarks may be attributed to several factors. (1) **Alignment of visual features with training data**: both tasks use math sketches, which are highly consistent with the GeoQA170K and Math PUMA datasets that constitute about 50% of our training samples. (2) **Nature of the tasks**: MathVerse (vision) and WeMath require explicit visual understanding and reasoning that go beyond pure language reasoning. Our VL-Thinker models are specifically designed and trained for such capabilities. For example, MathVerse (vision) involves identifying relationships among image components and reasoning over them, while WeMath comprises verified visual QA pairs requiring varied levels of knowledge and reasoning.
> We will add a sentence in the revised version to clarify these points: “The alignment between task-specific visual features and our training data, combined with the explicit visual reasoning demands of MathVerse and WeMath, explains why our models achieve greater improvements on these benchmarks.”
>
>
> > 4. Refine terminology for “Rule-based reward”.
>
> Thank you for highlighting the potential ambiguity in our terminology. We acknowledge that “rule-based reward” may not perfectly capture the intended meaning. At the same time, this phrasing has been adopted in several prior works [1,2,3] to describe verifiable rewards used for RL training. Therefore, we chose to follow the same terminology for consistency with the literature. Nevertheless, we appreciate your suggestion and would be grateful if you could recommend an alternative phrasing that would better convey the idea.
>
> [1] Meng, Fanqing, et al. "Mm-eureka: Exploring the frontiers of multimodal reasoning with rule-based reinforcement learning." arXiv preprint arXiv:2503.07365 (2025).
>
> [2] Guo, Daya, et al. "Deepseek-r1: Incentivizing reasoning capability in llms via reinforcement learning." arXiv preprint arXiv:2501.12948 (2025).
>
> [3] Shen, Haozhan, et al. "Vlm-r1: A stable and generalizable r1-style large vision-language model." arXiv preprint arXiv:2504.07615 (2025).
>
> > 5. Introducing MCQ; color schemes in fig3 and 5; marking best performance in tab4.
>
> “MCQ” stands for “Multiple Choice Questions”. We will fix these design and formatting issues in the updated version!

---

### Author Response · Authors · 2025-09-11
**General Response**

We thank all reviewers for their valuable inputs to improve our work!

1. **Refined conclusion**: Reviewer M1xC and 6XCA both mention that the conclusion should be stated with stricter constraints given our experiment setting. Therefore, we narrow our conclusion as: **on vision-language math reasoning tasks, we find degraded performance of Qwen-2/2.5VL models when reasoning SFT is applied, while GRPO can substantially improve their performance**. The refined conclusion provides insights on training Qwen2/2.5VL models (which are one of the most popular choices of VL backbones [1,2,3,4,5]) for VL math reasoning tasks (which is one of the key task families to measure model’s reasoning ability). Albeit being narrower than the original version, this conclusion is better supported by our experiment results and fills the gap of previous research.

    [1] Meng, Fanqing, et al. "Mm-eureka: Exploring the frontiers of multimodal reasoning with rule-based reinforcement learning." arXiv preprint arXiv:2503.07365 (2025).

    [2] Shen, Haozhan, et al. "Vlm-r1: A stable and generalizable r1-style large vision-language model." arXiv preprint arXiv:2504.07615 (2025).

   [3] Huang, Wenxuan, et al. "Vision-r1: Incentivizing reasoning capability in multimodal large language models." arXiv preprint arXiv:2503.06749 (2025).

   [4] Wang, Haozhe, et al. "Vl-rethinker: Incentivizing self-reflection of vision-language models with reinforcement learning." arXiv preprint arXiv:2504.08837 (2025).

   [5] Ouyang, Kun, et al. "SpaceR: Reinforcing MLLMs in Video Spatial Reasoning." arXiv preprint arXiv:2504.01805 (2025).



2. **Evaluation of training set quality**:
Reviewer 6XCA and 7XcF both request a quality check on the training set distilled from R1.
We evaluate our SFT data quality using GPT-5 on 100 randomly sampled examples (uniformly sampled from each data source) across 7 error dimensions. Our prompt is as follows:
```
You are an expert evaluator tasked with assessing the quality of reasoning traces in visual question answering tasks. Your job is to identify specific types of errors in the reasoning process.

**TASK**: Evaluate the following reasoning trace for a visual question answering problem.

**QUESTION**: {question}

**REASONING TRACE**:
{reasoning_trace}

**EVALUATION CRITERIA**:
Please evaluate the reasoning trace and count the number of errors in each of the following categories:

1. **FACTUAL_ERRORS**: Incorrect statements about the image content, object properties, or visual details
2. **LOGICAL_ERRORS**: Flawed logical reasoning, incorrect mathematical operations, or invalid deductions
3. **INCOMPLETE_REASONING**: Missing steps, incomplete analysis, or failure to address all aspects of the question
4. **INCONSISTENCY_ERRORS**: Contradictory statements within the reasoning trace
5. **HALLUCINATION_ERRORS**: Describing objects, properties, or details that are not present in the image
6. **PROCESSING_ERRORS**: Errors in understanding the question, misinterpreting instructions, or wrong problem formulation
7. **VERIFICATION_ERRORS**: Failure to double-check calculations, assumptions, or conclusions

**INSTRUCTIONS**:
- Carefully read through the entire reasoning trace
- Identify specific instances of each error type
- Count the total number of errors in each category
- Be precise and conservative in your evaluation
- If no errors of a particular type are found, return 0 for that category
- Focus on the reasoning process, not just the final answer

**OUTPUT FORMAT**:
Return your evaluation as a JSON object with the following structure:
{{
    "factual_errors": <number>,
    "logical_errors": <number>,
    "incomplete_reasoning": <number>,
    "inconsistency_errors": <number>,
    "hallucination_errors": <number>,
    "processing_errors": <number>,
    "verification_errors": <number>,
    "overall_quality_score": <score from 1-10>,
    "explanation": "<brief explanation of the main issues found>"
}}

**IMPORTANT**:
- Return ONLY the JSON object, no additional text
- Ensure the JSON is valid and properly formatted
- The overall_quality_score should be from 1 (very poor) to 10 (excellent)
- Be objective and fair in your assessment
```

**Our analysis reveals that data quality is NOT the primary issue:**
- Overall quality: 7.98 out of 10
- Factual errors: Only 2% of samples
- Mathematical questions: 8.78/10 with 84.4% high quality

The low error rates demonstrate that our caption+QA approach for the text-only R1 effectively transfers visual information. The performance gap between SFT and RL stems from fundamental differences in learning paradigms: SFT teaches format imitation while RL enables genuine reasoning through exploration and feedback. We will add this result to our camera-ready version.

---

### Decision · Action_Editor_zGp6 · 2025-10-07

**Recommendation:** Accept with minor revision

**Additional Comments:**

Overall, the reviewers were more than satisfied with the revised version of the paper and I concur that this paper presents interesting findings that can stimulate further discussions. I request the authors to take the reviewers concerns into consideration while preparing the final version of the paper.

Congratulations on the acceptance.

**Audience:**

Yes

**Audience Explanation:**

The research area of reasoning capabilities of VLMs will be of widespread interest to the TMLR audience.

**Claims And Evidence:**

Yes

**Claims Explanation:**

The work sets out to answer two central questions prevelant in developing reasoning abilities in Large Vision Language Models, namely:
* What are the distinct effects of supervised fine tuning (SFT) and reinforcement learning (RL) in multimodal reasoning? and
* Is this two-stage paradigm absolutely necessary for reasoning in VLMs?

 The authors construct a large-scale dataset suitable for both SFT and RL using an automated data collection pipeline and put the SFT-then-RL paradigm used to traion VLMs to the test. The authors conduct experiments comparing SFT, RL and combination of both using Qwen2VL, Qwen2.5VL on various visual math reasoning benchmarks. The final result of these experiments is that RL is a significantly more effective strategy than SFT for training reasoning VLMs.

The tackled problem is very intereting and are backed up by sufficient empirical evidence. Reasoning in large models is still a nascent are of research and this paper sheds light on a more fundamental part of the reasoning process, i.e.e the training procedure of such a model.

The reviewers raised several initial questions such as lack of common sense reasoning benchmarks, framing of the findings of the paper being too broad and the choice of the underlying VLMs being arbitrary. The rebuttal phase cleared most of the concerns and the reviewers were more supportive of the paper.